# THE CHICKEN AND EGG DILEMMA: CO-OPTIMIZING DATA AND MODEL CONFIGURATIONS FOR LLMS

## ABSTRACT

Co-optimizing data and model configurations for LLMs presents a classic chicken-and-egg dilemma: the best training data configuration (e.g., training data composition) depends on the chosen model configuration (e.g., model architecture, fine-tuning configuration), but the best model configuration also depends on the chosen training data. However, jointly optimizing both data and model configurations is intractable, with existing methods focusing only on data or model selection in isolation without considering their complex interdependence. We introduce `JoBS`, an efficient method that *jointly* optimizes LLM training data and model configurations by framing the problem as a black-box optimization problem. Central to our method is a novel performance scaling law predictor, which learns a diverse family of performance scaling laws for different configurations and cheaply predicts how promising a particular training configuration is. This enables us to quickly build an approximate LLM performance landscape and efficiently find optimal training configurations with Bayesian Optimization (BO). `JoBS` not only outperforms existing baselines across diverse tasks in the fine-tuning setting, but also runs up to 12.4× faster. We hope our work draws more attention to the chicken-and-egg dilemma inherent in co-optimizing LLM training configurations. Our anonymized code is available at: `https://github.com/a35453779/JoBS`.

## 1 INTRODUCTION

LLMs have become ubiquitous in our lives, with great commercial and practical interest in maximizing their performance for specific tasks. Much effort has been put into optimizing the *training components* to maximize LLM performance, particularly the *training data* and the *model architecture*. From the data perspective, better training data can be chosen via data selection (Koh & Liang, 2020; Xie et al., 2023b; Xia et al., 2024; Chen et al., 2025c) and mixing (Xie et al., 2023a; Chen et al., 2025a;c; Liu et al., 2025; Xie et al., 2025) techniques. From the model perspective, various model selection methods (Raschka, 2020; White et al., 2020; He et al., 2024; Zhang et al., 2024b) have been introduced to select the most appropriate model for a given task.

In practice, optimizing training data and model architecture is a highly interdependent process. For example, deploying data selection methods requires us to first assume a good model architecture. Conversely, selecting a good model architecture requires a fixed pool of training data. This presents a classic *chicken-and-egg dilemma*, where the optimal choice of training data depends on the optimal choice of model architecture, and vice versa. Furthermore, due to their interdependency, optimizing data and model *independently* would often lead to sub-optimal LLM performance (Chen et al., 2024). This is demonstrated in Sec. 5 where we naively combined data and model selection methods. Therefore, to address the interdependent nature of data and model architecture and maximize LLM performance, we should *jointly* optimize these two training components.

Unfortunately, jointly optimizing data and model configurations is conventionally considered challenging and budget-intensive. Prior scaling law works (Kaplan et al., 2020; Hoffmann et al., 2022; Zhang et al., 2024a; Shukor et al., 2025) have tried to quantify the effects of each training component on downstream performance, while prescribing simple guidelines on the optimal choices of training components given fixed computational budgets. However, they require exhaustive search over a large number of configurations, which is infeasible in practice. To *efficiently* find an optimal training configuration is therefore a problem that remains difficult and largely unexplored.

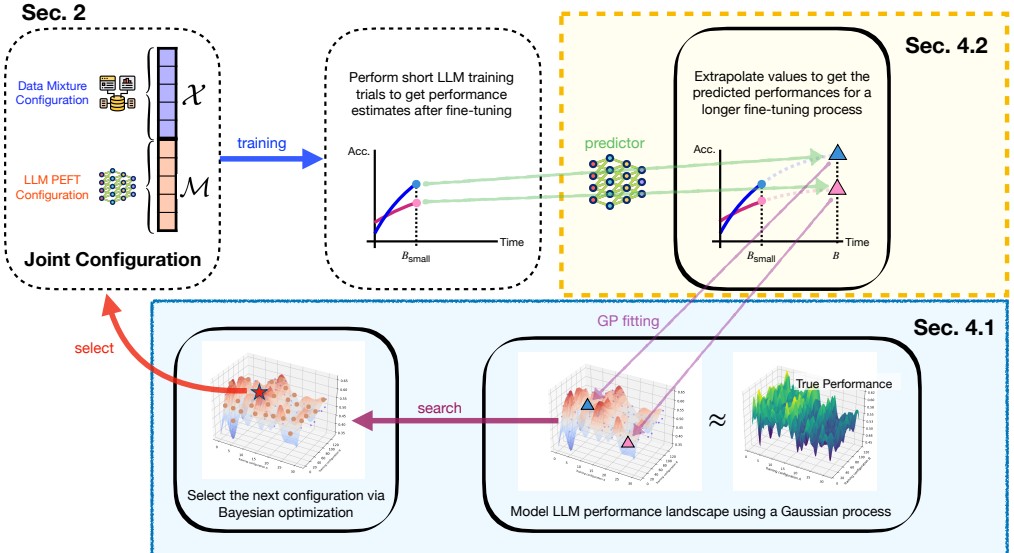

Figure 1: Overview of JoBS.

Our paper aims to study this chicken-and-egg dilemma and joint optimization problem for a scenario commonly faced by practitioners, namely *parameter-efficient fine-tuning* (Hu et al., 2021) (PEFT) of LLMs under different *data mixtures*. In this work, we present **Jo**int **B**ayesian Optimization with **S**caling Laws (JoBS), an approach that efficiently co-optimizes LLM training configurations by learning the LLM performance landscape with Bayesian Optimization (BO) and a novel performance predictor to reduce amount of actual training. We offer both theoretical and empirical insights into how fine-tuning performance varies with different *Low Rank Adaptation* (LoRA) configurations and training data mixture choices. In doing so, we address the research gap in studying the complex interaction between data and model configurations and jointly optimize both components efficiently. We summarize JoBS in Fig. 1, and state our main contributions below:

1. We formulate our chicken-and-egg dilemma as a black-box optimization problem (Sec. 2) and provide novel empirical and theoretical insights into how choices of LoRA configuration and training data mixture jointly influence the LLM fine-tuning performance (Sec. 3). Our work is the first to explore and quantify the interaction gains from co-optimizing model and data configurations for an LLM. We find that the LLM *performance landscape* is *approximately smooth* with respect to varying training configurations, and good configurations can improve LLM performance by more than 20%.

2. We present JoBS (Sec. 4), an algorithm that exploits the discovered characteristics of the co-optimization problem, and interleaves *Bayesian Optimization* (BO) (Sec. 4.1) with a novel LLM performance scaling law predictor to efficiently learn the smooth performance landscape (Sec. 4.2). The predictor effectively amortizes expensive trials in BO, allowing us to efficiently co-optimize training configuration – a traditionally costly endeavor – with theoretical performance guarantees.

3. We empirically demonstrate the performance gains attained by JoBS (Sec. 5). By comparing our algorithm with a wide range of independent model and data selection baselines, we show the existence of an *interaction improvement* – a nugget of performance improvement from co-optimizing the training configurations, which is a $6 - 7\%$ performance increase compared to merely optimizing each training component independently.

## 2 PROBLEM SETUP AND RELATED WORKS

We consider two categories of training components: **training data** $\mathcal{X}$ and **model** $\mathcal{M}$. Given these training components, we define a training process $P_t$ that fine-tunes an LLM for a training time of $t$ to produce fine-tuned LLM weights $\theta_{\mathcal{X}, \mathcal{M}, t} \triangleq P_t(\mathcal{X}, \mathcal{M})$, which can be evaluated over a predefined

performance metric $\mathcal{L}$ (e.g., question-answering accuracy). Given a training time budget $B$, we want to find training component configurations $\mathcal{X}, \mathcal{M}$ that maximize the LLM performance metric:

$$\max_{\mathcal{X},\mathcal{M}} \mathcal{L}(\theta_{\mathcal{X},\mathcal{M},B}). \tag{1}$$

Time budget $B$ is considered since in practice, a model cannot be trained indefinitely. As different training configurations have different training speeds, the time budget forces us to strategically balance between each training component to attain the best LLM performance within a practical resource constraint. Other constraints, such as training tokens, are correlated with training time and can also be considered, but we find training time easier for practitioners to interpret.

**Data $\mathcal{X}$.** Assume we have $N$ training datasets $\mathcal{D} \triangleq D_1 \cup D_2 \cup \cdots \cup D_N$ from $N$ different domains (e.g., Wikipedia, TruthfulQA (Lin et al., 2022) for language tasks). The training data component consists of a subset of data $\mathcal{X} \subseteq \mathcal{D}$. In general, the selection of $\mathcal{X}$ ensures the selected data points are more relevant to the given task (Chen et al., 2025c) or of higher quality (Wang et al., 2024a; Xia et al., 2024; Zhang et al., 2025), however this is done so assuming a fixed model architecture is used. In our work, we overload the notation and parameterize our selected data mixture with a mixing ratio represented by a probability simplex of dimension $N$ ($\mathcal{X} \in \Delta^{N-1} \subset \mathbb{R}^N$).

**Model $\mathcal{M}$.** Under the LLM PEFT regime, the optimization problem takes as inputs: (1) the *module* of the LLM to which PEFT is applied (e.g., $Q, V$ projection (Vaswani et al., 2017)), (2) the *layer(s)* where PEFT is applied (e.g., layer 30), and (3) the *PEFT hyperparameters*, including LoRA rank, $\alpha$ and dropout (Hu et al., 2021). These inputs can be concatenated into a $M$-dimensional vector $\mathcal{M} \in \mathbb{R}^M$ with $M \in \mathbb{Z}^+$. These inputs span both discrete and continuous spaces, which complicates the optimization problem. Existing model selection works (Raschka, 2020; White et al., 2020; He et al., 2024; Zhang et al., 2024b) can be adapted to select configurations for PEFT, however these methods assume that a fixed training dataset is known beforehand.

## 3 Motivation for JoBS

Solving Problem 1 directly is challenging. This is because the performance landscape that describes the relationship between selected training components $\mathcal{X}, \mathcal{M}$ and the fine-tuned LLM performance $\mathcal{L}$ has no closed, analytical form. Before introducing JoBS as an efficient approach, we first examine how different training data and model configurations shape the LLM performance landscape. These findings are counter-intuitive yet illustrative, giving us a clearer understanding of the LLM performance landscape and justifying why our chicken-and-egg dilemma deserves attention in the first place. We will use these findings to motivate the algorithmic backbone of JoBS later in Sec. 4.

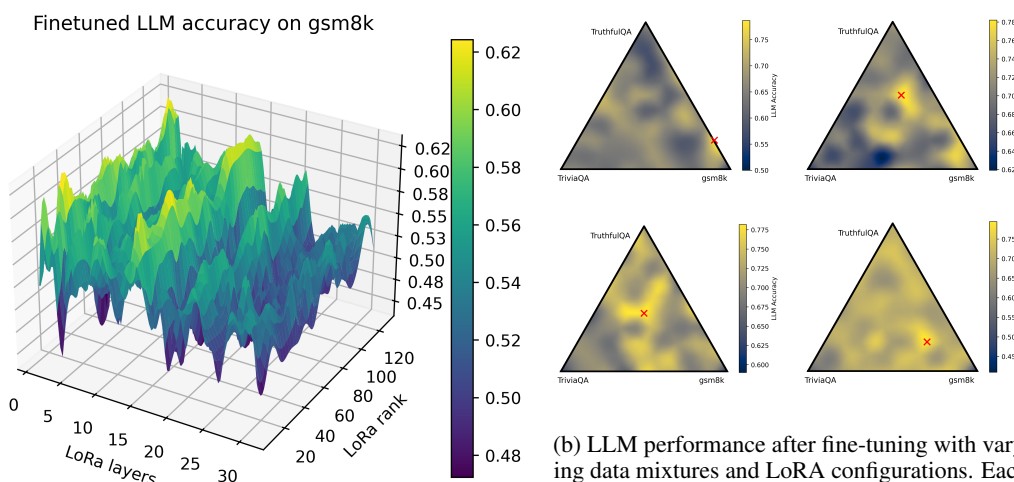

(a) LLM performance after fine-tuning with LoRA applied on varying number of layers and with varying LoRA rank.

(b) LLM performance after fine-tuning with varying data mixtures and LoRA configurations. Each chart corresponds to a different LoRA configuration, while the red cross denotes optimal mixture for that LoRA configuration.

Figure 2: How data and model configurations jointly affect LLM performance.

**Model configurations significantly influence downstream performances.** To demonstrate this, we fine-tuned a Llama-3-8B-Instruct (Touvron et al., 2023) model on the gsm8k (Cobbe et al., 2021)

task with LoRA (Hu et al., 2021) with different ranks and layers[1]. For each LoRA rank and layer configuration, we fine-tuned the model for one epoch. Intuitively, we expect the LLM to perform better if we use LoRA with larger ranks and applied to more layers, due to a higher learning capacity. Surprisingly, this is not the case. We instead found that the performance landscape is somewhat smooth but riddled with "peaks" and "valleys", and certain LoRA layer and rank configurations yield *drastically* better performance (almost 20%!) than simply fine-tuning over all model layers and larger ranks. Unlike many practical works that merely prescribe a LoRA configuration from heuristics, our finding suggests that certain LLM model configurations produce far better LLM performance, and we should optimize them while considering the chicken-and-egg dilemma. Beyond just LoRA rank and layers considered in prior works (Zhang et al., 2024b), our paper also considered other model configurations such as *which LLM modules* to apply LoRA to and more (Sec. 5.1).

**The optimal data configuration varies between chosen model configuration.** To demonstrate this, we fine-tuned an LLM with varying data mixtures for the gsm8k (Cobbe et al., 2021) evaluation task whilst varying the model configuration in which we applied LoRA. Our training data mixture consists of 3 training domains: TruthfulQA (Lin et al., 2022), TriviaQA (Joshi et al., 2017) and gsm8k. Intuitively, we expect the LLM to perform best if we only used training data from gsm8k. However, this is not the case; Fig. 2b shows that the optimal data mixture (red cross) contains a mixture of data points from different domains. This suggests that the optimal training data mixture is non-intuitive and difficult to find via heuristics (Radford et al., 2019; Gao et al., 2020). More importantly, the optimal training data mixture seems to vary with different model configurations, yielding varying LLM performance. Therefore, these preliminary results emphasize the need to derive algorithms to *automatically* and *jointly* optimize all training components.

Lastly, we refer interested readers to some theoretical insights that we developed from classical convex optimization in App. A, which helps us understand the optimal training configuration choice.

## 4 INTRODUCING JOBS

JOBS features two main components. (1) We use a surrogate Gaussian process (Williams & Rasmussen, 2006) to model the empirically smooth performance function landscape $\mathcal{L}$ (shown earlier), whose maximum can be obtained in a sample-efficient manner by Bayesian optimization (Sec. 4.1). (2) We introduce a novel performance scaling law (Wu & Tang, 2024; Chen et al., 2025b) predictor that amortizes the repeated cost of repeated evaluations by estimating the LLM performance from a small number of training steps (Sec. 4.2). Unlike existing rigid scaling law formulas which are fixed to a small group of training configurations, our predictor is a flexible neural network, capable of predicting LLM performance scaling w.r.t. *any training configurations*.

We show theoretically (in Sec. 4.3) and empirically (in Sec. 5) that even when our LLM performance predictions are noisy, the BO framework handles them gracefully as *observation noise*, eventually converging to the optimal training component configuration.

### 4.1 BO AS THE BACKBONE OF JOBS

**Black-box modeling of the trained LLM performance.** We consider the LLM performance as a function $\mathcal{L} : \mathbb{R}^d \mapsto \mathbb{R}$ over the space of inputs $x = [\mathcal{X}, \mathcal{M}] \in \mathbb{R}^d$ where $d = N + M$ (See Sec. 2). Since it is difficult to analytically model the LLM performance $\mathcal{L}$, we instead treat our objective function in Problem 1 as a *black-box function* whose maximum $x^* \triangleq \text{argmax}_x \mathcal{L}(x)$ we want to recover. In line with existing works, we attempt to model $\mathcal{L}$ as a *Gaussian process* (GP) (Williams & Rasmussen, 2006). In each iteration $t = 1, 2, \ldots, T$, we can trial some training configuration $x_t$ to obtain a potentially *noisy* realization of the LLM performance $y_t \triangleq \mathcal{L}(x_t) + \epsilon_t$, which we assume is corrupted with a sub-Gaussian noise $\epsilon_t$ (e.g., Gaussian or bounded noise) to form the sample $(x_t, y_t)$.

Consistent with the work of Chowdhury & Gopalan (2017), we model the unknown function $\mathcal{L}$ (in our case, the LLM performance landscape) as a realization of a GP that is fully specified by its *prior* mean $\mu(r)$ and covariance $\kappa(x, x')$ for all $x, x' \in \mathbb{R}^d$ where $\kappa$ is a *kernel* function chosen to characterize the correlation of the observations between any two inputs $x$ and $x'$. For JOBS, since we expect the

---

[1]Generating this simple performance landscape took a few weeks, so exhaustively searching for the optimal configuration is infeasible.

function $\mathcal{L}$ to be heteroskedastic and have varying lengthscales between different inputs, we use a deep kernel (Wilson et al., 2016) which provides greater modeling flexibility. The hyperparameters in the mean and kernel functions can be learned via maximum likelihood estimation from observations.

Given the noisy observations $\boldsymbol{y}_t \triangleq [y_\tau]_{\tau=1,\dots,t}^\top$ at inputs $x_1, \dots, x_t$, the posterior belief of $\mathcal{L}$ at any new input $x'$ is a Gaussian distribution with the *posterior* mean and variance given by

$$\mu_t(x') \triangleq \kappa_t^\top(x')(K_t + \zeta I)^{-1}\boldsymbol{y}_t$$
$$\sigma_t(x') \triangleq \kappa(x', x') - \kappa_t^\top(x')(K_t + \zeta I)^{-1}\kappa_t(x') \tag{2}$$

where $\kappa_t(x') \triangleq [\kappa(x', x_\tau)]_{\tau=1,\dots,t}^\top$ is a column vector, $K_t \triangleq [\kappa(x_\tau, x_{\tau'})]_{\tau,\tau'\in 1,\dots,t}$ is a $t \times t$ covariance matrix, and $\zeta > 0$ is viewed as a free hyperparameter (Chowdhury & Gopalan, 2017). Modeling $\mathcal{L}$ directly allows the entire performance landscape to be learned at once, as opposed to learning a slice of $\mathcal{L}$ for a fixed $\mathcal{X}$ or $\mathcal{M}$. This results in more efficient learning process and avoiding heuristics to balance between which $\mathcal{X}$ or $\mathcal{M}$ to trial, making JoBS more robust overall.

**Using BO for our joint optimization problem.** To determine the best configuration $x^*$, we trial different training configurations in each round to determine their performance and continually update the GP in (2) to have a better estimate of $\mathcal{L}$. In round $t$, the BO algorithm proposes the next configuration $x_{t+1}$ as the configuration which maximizes some acquisition function, such as the *upper confidence bound* (UCB) (Srinivas et al., 2010), given by $x_{t+1} = \operatorname{argmax}_x \mu_t(x) + \beta_{t+1}\sigma_t(x)$, where $\beta_{t+1}$ is an exploration parameter which decays with increasing $t$. We can assess the convergence of a BO algorithm by analyzing its cumulative regret after $T$ BO iterations, given by $R_T \triangleq \sum_{t=1}^T [\mathcal{L}(x^*) - \mathcal{L}(x_t)]$ (Tay et al., 2023), where $\mathcal{L}(x^*)$ is the optimum. A lower cumulative regret indicates a faster convergence rate of the BO algorithm. We provide a theoretical analysis of JoBS's cumulative regret in Sec. 4.3.

We outline a few practical methods to improve BO in our problem setting. *First*, we use the constrained BO formulation (Eriksson & Poloczek, 2021; Chen et al., 2025c) to constrain the sum of data mixture ratio in our data configuration $\mathcal{X}$ to 1. *Second*, a number of our problem inputs is discrete in nature (e.g., whether to apply LoRA to the LLM Q-projection layer, see Sec. 5.1). To address this, we adopt continuous parameterization (Daulton et al., 2022) to perform BO effectively over a mixture of such discrete and continuous input spaces.

## 4.2 USING PERFORMANCE PREDICTOR TO IMPROVE COMPUTATION TIME

While BO searches through different training configurations in a sample-efficient manner (Srinivas et al., 2010) and avoids performing exhaustive search over all possible $x$, naively applying BO still requires lengthy fine-tuning in each iteration. For example, if $B = 1000$s, we need to fine-tune for 1000 seconds in each round. To speed up JoBS, we take inspiration from LLM performance scaling laws (Wu & Tang, 2024; Chen et al., 2025b) and introduce a novel performance predictor to estimate the full fine-tuning LLM performance from a shorter training trial (See Fig. 1).

For our predictor to work, we need to predict LLM performance for different training configurations (that we do not know in advance) at each BO iteration. Hence, we cannot use existing scaling laws (Kaplan et al., 2020; Wu & Tang, 2024; Chen et al., 2025b), which are defined with respect to a *fixed* training configuration. To address this issue, JoBS learns a neural network which takes *any* training configuration $[\mathcal{X}, \mathcal{M}]$ and its performance $\mathcal{L}(\theta_{\mathcal{X},\mathcal{M},B_{small}})$ at time $B_{small} < B$ as inputs and predicts the final fine-tuned LLM performance. Our predictor **does not** predict the full "scaling curve", but rather directly gives the performance after fine-tuning for time $B$.

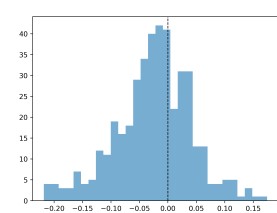

Figure 3: $\mathcal{F}$ prediction error.

JoBS learns this predictor in two steps. *First*, it collects a random Sobol sequence (Nguyen et al., 2018) of initial training configurations in $\mathcal{X}, \mathcal{M}$ and observe LLM performance at small time step $\mathcal{L}(\theta_{\mathcal{X},\mathcal{M},B_{small}})$ and large time step $\mathcal{L}(\theta_{\mathcal{X},\mathcal{M},B})$. These observations are also used to fit our GP to approximate our performance landscape (Sec. 4.1), and therefore are not wasted after the predictor has been trained. *Second*, using the observations, it fits a predictor neural network $\mathcal{F} : \mathcal{X}, \mathcal{M}, \mathcal{L}(\theta_{\mathcal{X},\mathcal{M},B_{small}}) \mapsto \mathcal{L}(\theta_{\mathcal{X},\mathcal{M},B})$ that extrapolates how well an LLM performs from a small amount of training time $B_{small}$. We provide examples of the extrapolation learnt by our predictor $\mathcal{F}$

in Fig. 4d and its prediction error in Fig. 3. If available, we can also use prior performance reported from past experiments or papers to accelerate the neural network training.

At each step of JoBS, we only fine-tune the LLM for time $B_{\text{small}}$ to observe $\mathcal{L}(\theta_{\mathcal{X},\mathcal{M},B_{\text{small}}})$, then use $\mathcal{F}$ to estimate the full fine-tuning performance $\hat{\mathcal{L}}(\theta_{\mathcal{X},\mathcal{M},B})$. These cheap estimates effectively allow JoBS to learn the performance landscape without fine-tuning the LLM to completion.

### 4.3 CONVERGENCE UNDER PRESENCE OF PREDICTION NOISE

We have amortized and reduced the runtime of JoBS by predicting the LLM performance $\mathcal{L}(\theta_{\mathcal{X},\mathcal{M},B})$ of a particular training configuration. However, we obviously cannot make perfect predictions. As such, we can only observe $\hat{\mathcal{L}}(\theta_{\mathcal{X},\mathcal{M},B}) = \mathcal{L}(\theta_{\mathcal{X},\mathcal{M},B}) + \epsilon$ at each BO iteration, where $\epsilon$ is the prediction error associated with our predictor $\mathcal{F}$ introduced earlier (See Fig. 3). How does this prediction error influence the effectiveness of JoBS? We show that under some mild assumption on prediction error $\epsilon$ (as long as it is not too large), JoBS converges to the optimal training configuration with the following convergence rate. In other words, our predictor's error is handled gracefully by JoBS's BO backbone, allowing us to still find optimal configurations.

**Theorem 4.1.** *Let $\mathcal{L}(\theta_{\mathcal{X},\mathcal{M},B})$ be the performance landscape of training configuration with bounded RKHS norm: $\|\mathcal{L}\|_{\kappa} = \sqrt{\langle \mathcal{L}, \mathcal{L} \rangle_{\kappa}} \leq B$ w.r.t. kernel $\kappa$. Also, let $\gamma_T$ be the maximum information gain from $T$ iterations. As mentioned above, assume we make noisy observation $\hat{\mathcal{L}}(\theta_{\mathcal{X},\mathcal{M},B}) = \mathcal{L}(\theta_{\mathcal{X},\mathcal{M},B}) + \epsilon$ at each BO iteration and error $\epsilon$ associated with our scaling law prediction is Sub-Gaussian with a factor of $R$. Then, running our BO algorithm over training configurations $\mathcal{X}, \mathcal{M}$ with the IGP-UCB acquisition function (Chowdhury & Gopalan, 2017) yields the following cumulative regret with probability at least $1 - \delta$:*

$$R_T = \mathcal{O}\left( B\sqrt{T\gamma_T} + R\sqrt{T}\sqrt{\gamma_T^2 + \gamma_T \ln(1/\delta)} \right) \tag{3}$$

The proof is provided in App. B and shows that the prediction error of $\mathcal{F}$ in JoBS can be viewed as observation noise under the BO framework, allowing us to still uncover the optimal training configuration with sufficient BO iterations. **Our theoretical finding also uncovers an interesting compute-performance tradeoff**: extrapolating from a smaller amount of training time $B_{\text{small}}$ reduces the training time at each BO iteration, but incurs noisier prediction errors with larger $R$ constants, leading to larger cumulative regret. In Sec. 5.4, we examine how varying prediction errors from our performance predictor (adjusted with $B_{\text{small}}$) influence our algorithm's convergence.

## 5 EXPERIMENTS

We use JoBS to jointly optimize training configurations for LLM fine-tuning in a variety of language tasks and LLM model types. First, we show that when data and model selection methods are applied independently (or in an alternating manner) to LLM model and data configurations, they do not perform as well as JoBS because the former does not consider interactions between data and model configurations. Next, we make several interesting observations regarding JoBS's convergence w.r.t. different factors, such as the choice of $B_{\text{small}}$. Lastly, we perform a few ablations to tease apart the influence of different components in JoBS.

### 5.1 EXPERIMENTAL SETTINGS

In all our experiments, we aim to fine-tune an LLM for a fixed training budget to maximize its performance on an evaluation task. To make the task more difficult, *we adopt an out-of-domain setting* (Chen et al., 2025c), where the evaluation task's data is removed from the training domains. We use a data mixture from 10 datasets: **Wikitext** (Merity et al., 2016), **gsm8k** (Cobbe et al., 2021), **PubmedQA** (Jin et al., 2019), **HeadQA** (Vilares & Gómez-Rodríguez, 2019) , **SciQ** (Welbl et al., 2017), **TriviaQA** (Joshi et al., 2017), **TruthfulQA** (Lin et al., 2022), **MMLU** (Hendrycks et al., 2021), **AI2 ARC** (Clark et al., 2018) and **CommonsenseQA** (Talmor et al., 2019). We mix the datasets (Chen et al., 2025c; Xie et al., 2023a; Ye et al., 2024) to create a fine-tuning dataset consisting 10000 data points and consider the mixing ratio (a probability simplex) across these datasets as the training data configuration $\mathcal{X}$. The model configurations $\mathcal{M}$ we consider here are which LLM layer to apply LoRA to, which LLM module to apply LoRA to (e.g., Q projection), LoRA rank, LoRA

dropout and alpha, **giving us a total of 19 training configuration dimension**. Unless otherwise stated, we used 100 BO iterations for JoBS with $B_{small} = 50$ seconds, $B = 1000$ seconds and a batch size of 8. There are minor differences in our LLM performance from existing papers due to evaluation setup. More information on our experimental setup is provided in App. C.

## 5.2 BASELINES

**Data selection. LESS** (Xia et al., 2024) searches for more relevant data points based on their training gradients. **DoReMi** (Xie et al., 2023a) adopts a distributionally robust approach to produce data-mixtures that work generally well against every distribution of evaluation task. Influence Function (**IF**) (Koh & Liang, 2020) selects data points with the higher influence scores. **Diversity** (Wang et al., 2024b) finds the subset of data points with the largest log-determinant score. **BO** just performs vanilla BO on the data configuration.

**Model selection.** We used a variant of Differentiable Architecture Search (**DARTS**) (Liu et al., 2019) applied to our LoRA weights by tuning an additional mixture coefficient on each LLM layer (so, when this coefficient approaches zero for a layer, it implies we do not apply LoRA weight to that LLM layer). **AutoLoRA** (Zhang et al., 2024b) is a baseline that automatically tunes the LoRA rank, but does not consider how we should select the layers to apply LoRA to. **RoBoT** (He et al., 2024) adopts a training-free approach towards selecting different model configurations by aggregating different training-free metrics to measure how promising a given configuration is. **BO** just performs vanilla BO on the model configuration.

**Mix and match**. There are two ways to combine the baselines to ensure a good coverage of empirical comparison: we can either perform data and model selection independently in a one-shot setting or repeat them in an alternating manner using the current best-found model or data (e.g., optimize the model, then optimize the data, before repeating the process;). We performed the one-shot optimization approach in Table 1 and the alternating approach in Table 3. In both cases, they do not perform as well as JoBS. Roughly speaking, alternating between model and data selection is similar to coordinate descent (Wright, 2015) but does not guarantee optimality. We also explored other naive approaches (App. F.1), such as randomly choosing training configurations or only performing BO over model or data configurations, but found their performances lackluster.

## 5.3 MAIN RESULTS AND KEY TAKEAWAYS

In Sec. 4, we claimed that JoBS models the complex interaction between training components, jointly optimizing them effectively to attain better LLM performance. To verify this hypothesis, we mixed and matched conventional data selection and model architecture search methods and applied them to each training component independently. We compared this with JoBS, which jointly optimizes both training components. Due to space constraints, we only display the partial results for **gsm8k** and **TruthfulQA** here. Our results over other tasks are shown in App. F.

Table 1: **Evaluation task: gsm8k** (Cobbe et al., 2021) . Combination matrix of mixing and matching different model and data selection methods on LLM performance compared to our joint optimization approach (JoBS). Subscript numbers represent standard deviations across 5 trials. Due to space constraints, we show the results of other tasks in App. F

| ↓ **Model | Data** → | Default | LESS | DoReMi | IF | Diversity | BO | JoBS |
|---|---|---|---|---|---|---|---|
| Default | $68.1_{\pm 2.1}$ | $70.4_{\pm 1.1}$ | $71.6_{\pm 3.1}$ | $67.9_{\pm 0.9}$ | $73.8_{\pm 1.8}$ | $73.4_{\pm 1.7}$ | - |
| DARTS | $72.4_{\pm 0.8}$ | $71.0_{\pm 0.6}$ | $74.1_{\pm 1.3}$ | $68.7_{\pm 0.4}$ | $66.1_{\pm 0.7}$ | $72.8_{\pm 0.3}$ | - |
| AutoLoRA | $72.3_{\pm 1.1}$ | $74.6_{\pm 0.3}$ | $70.3_{\pm 0.7}$ | $67.9_{\pm 0.4}$ | $73.4_{\pm 0.5}$ | $72.5_{\pm 0.5}$ | - |
| RoBoT | $71.1_{\pm 0.6}$ | $72.0_{\pm 1.5}$ | $73.4_{\pm 1.8}$ | $72.4_{\pm 1.5}$ | $69.6_{\pm 1.7}$ | $72.4_{\pm 0.8}$ | - |
| BO | $70.7_{\pm 1.4}$ | $66.7_{\pm 0.8}$ | $72.5_{\pm 0.8}$ | $71.7_{\pm 0.9}$ | $74.7_{\pm 1.0}$ | $72.7_{\pm 2.3}$ | - |
| JoBS | - | - | - | - | - | - | $\mathbf{80.4}_{\pm 1.9}$ |

**Pairing different data and model selection methods (Table 1, 2 and App. F)**. Our results in the combination matrix showcase the shortfall of simply combining different model and data selection method. Simply pairing these methods independently does not consider the interaction between data and model configurations together, yielding mediocre performance. In contrast, JoBS attains higher

Table 2: **Evaluation task: TruthfulQA** (Lin et al., 2022).

| ↓ Model \| Data → | Default | LESS | DoReMi | IF | Diversity | BO | JoBS |
|---|---|---|---|---|---|---|---|
| Default | $55.4_{\pm 1.6}$ | $56.4_{\pm 0.8}$ | $58.2_{\pm 2.4}$ | $57.3_{\pm 1.1}$ | $59.8_{\pm 1.0}$ | $70.2_{\pm 0.8}$ | - |
| DARTS | $56.7_{\pm 1.1}$ | $57.0_{\pm 0.4}$ | $62.8_{\pm 1.1}$ | $59.1_{\pm 0.3}$ | $59.6_{\pm 1.0}$ | $72.4_{\pm 0.8}$ | - |
| AutoLoRA | $56.0_{\pm 0.8}$ | $62.6_{\pm 1.0}$ | $58.8_{\pm 0.9}$ | $59.6_{\pm 1.0}$ | $60.8_{\pm 0.4}$ | $68.4_{\pm 0.3}$ | - |
| RoBoT | $59.1_{\pm 0.4}$ | $60.2_{\pm 0.5}$ | $53.4_{\pm 1.1}$ | $52.4_{\pm 0.8}$ | $60.9_{\pm 0.4}$ | $69.6_{\pm 1.1}$ | - |
| BO | $61.0_{\pm 1.0}$ | $62.0_{\pm 0.3}$ | $64.0_{\pm 0.7}$ | $64.8_{\pm 0.8}$ | $60.3_{\pm 1.2}$ | $71.7_{\pm 1.8}$ | - |
| JoBS | - | - | - | - | - | - | $\mathbf{75.8}_{\pm 1.9}$ |

Table 3: Comparison of baselines with JoBS. Results are shown w.r.t. different evaluation tasks and LLMs (Higher is better), averaged over 5 trials. We choose to present a few better performing baselines (combining data and model selection methods in an alternating manner).

| Model | Task | Default fine-tuning | LESS + AutoLoRA | DoReMi + DARTS | Alternating-BO | JoBS |
|---|---|---|---|---|---|---|
| Llama-3-8B-Instruct | gsm8k | $68.1_{\pm 2.1}$ | $74.8_{\pm 0.9}$ | $73.2_{\pm 1.4}$ | $75.8_{\pm 1.8}$ | $80.4_{\pm 1.9}$ |
| | TruthfulQA | $55.4_{\pm 1.6}$ | $66.2_{\pm 0.7}$ | $68.9_{\pm 1.2}$ | $71.7_{\pm 1.1}$ | $75.8_{\pm 1.3}$ |
| | CommonsenseQA | $76.3_{\pm 1.0}$ | $80.5_{\pm 1.4}$ | $79.9_{\pm 1.0}$ | $78.5_{\pm 0.8}$ | $84.3_{\pm 2.4}$ |
| | HeadQA | $47.0_{\pm 0.9}$ | $46.5_{\pm 1.5}$ | $54.0_{\pm 1.8}$ | $56.3_{\pm 1.3}$ | $55.8_{\pm 1.5}$ |
| | MMLU | $61.2_{\pm 1.3}$ | $67.6_{\pm 2.9}$ | $64.1_{\pm 1.1}$ | $63.1_{\pm 1.1}$ | $69.5_{\pm 0.8}$ |
| | ARC | $54.7_{\pm 1.3}$ | $66.3_{\pm 1.6}$ | $62.5_{\pm 0.7}$ | $67.6_{\pm 0.6}$ | $70.4_{\pm 1.3}$ |
| | TriviaQA | $61.3_{\pm 2.4}$ | $70.4_{\pm 3.6}$ | $71.3_{\pm 1.4}$ | $74.6_{\pm 1.4}$ | $76.2_{\pm 1.2}$ |
| Qwen2.5-7B-Instruct | gsm8k | $70.2_{\pm 0.6}$ | $73.7_{\pm 0.9}$ | $71.1_{\pm 1.4}$ | $74.5_{\pm 3.1}$ | $81.3_{\pm 1.4}$ |
| | TruthfulQA | $56.4_{\pm 0.7}$ | $67.2_{\pm 1.3}$ | $68.3_{\pm 0.9}$ | $70.7_{\pm 0.8}$ | $74.8_{\pm 1.7}$ |
| | CommonsenseQA | $77.6_{\pm 0.4}$ | $82.1_{\pm 0.3}$ | $80.2_{\pm 0.6}$ | $80.6_{\pm 1.1}$ | $81.7_{\pm 0.6}$ |
| | HeadQA | $52.5_{\pm 0.3}$ | $51.3_{\pm 1.4}$ | $50.8_{\pm 0.9}$ | $54.5_{\pm 0.6}$ | $58.6_{\pm 0.9}$ |
| | MMLU | $72.5_{\pm 1.4}$ | $73.9_{\pm 1.6}$ | $72.8_{\pm 0.3}$ | $76.5_{\pm 1.2}$ | $78.4_{\pm 1.2}$ |
| | ARC | $64.6_{\pm 0.8}$ | $69.1_{\pm 3.1}$ | $71.5_{\pm 3.2}$ | $73.1_{\pm 1.1}$ | $75.0_{\pm 0.2}$ |
| | TriviaQA | $55.3_{\pm 0.8}$ | $65.1_{\pm 2.0}$ | $64.8_{\pm 1.2}$ | $62.0_{\pm 0.3}$ | $68.5_{\pm 1.3}$ |
| Mistral-7b-Instruct-v0.3 | gsm8k | $52.2_{\pm 0.8}$ | $58.7_{\pm 0.6}$ | $63.0_{\pm 1.1}$ | $62.2_{\pm 0.8}$ | $66.4._{\pm 0.5}$ |
| | TruthfulQA | $56.4_{\pm 0.7}$ | $59.8_{\pm 1.7}$ | $62.2_{\pm 0.6}$ | $69.4_{\pm 1.5}$ | $73.5_{\pm 0.6}$ |
| | CommonsenseQA | $77.6_{\pm 0.4}$ | $78.3_{\pm 1.1}$ | $77.9_{\pm 1.2}$ | $82.2_{\pm 0.7}$ | $83.5_{\pm 0.8}$ |
| | HeadQA | $57.8_{\pm 0.3}$ | $56.3_{\pm 0.9}$ | $57.9_{\pm 1.2}$ | $59.2_{\pm 1.1}$ | $57.8_{\pm 0.5}$ |
| | MMLU | $63.6_{\pm 0.5}$ | $71.8_{\pm 0.9}$ | $71.6_{\pm 1.3}$ | $72.3_{\pm 0.8}$ | $73.8_{\pm 0.9}$ |
| | ARC | $66.3_{\pm 0.8}$ | $70.2_{\pm 2.0}$ | $72.9_{\pm 1.0}$ | $72.4_{\pm 0.8}$ | $74.7_{\pm 0.6}$ |
| | TriviaQA | $58.2_{\pm 0.3}$ | $57.8_{\pm 1.8}$ | $60.5_{\pm 0.5}$ | $62.0_{\pm 0.3}$ | $66.3_{\pm 1.1}$ |

performance gains after fine-tuning, largely because it models and exploits the complex interaction between data and model configurations with the learnt performance landscape. By jointly optimizing both components, we attain a flat $6 - 7\%$ "interaction improvement" over other baselines.

**Alternating optimization scheme under same compute budget (Table 3).** Next, we selected a few better-performing optimization pairings from Table 1, 2 and *applied them in an alternating fashion to our training configurations* for 5 iterations. In general, data and model selection baselines are more computationally expensive, so this is a fair equal-compute comparison (See App. D). Table 3 shows that even when we ran data and model selection baselines in an alternating optimization scheme, the baselines do not perform as well as JoBS. In fact, for some tasks or models, the LLM performance of baselines becomes worse than that in the one-shot optimization scheme presented earlier. We speculate that this occurs because alternating optimization schemes might end up in worse-performing "saddle points" in the performance landscape, leading to performance degradation.

## 5.4 ABLATION AND ADDITIONAL ANALYSIS

In the previous sections, we showed that JoBS outperforms baselines in a variety of evaluation tasks. However, several questions remain regarding the performance-compute tradeoff in JoBS. For instance, how does our neural network predictor (Sec. 4.2) and $B_{small}$ affect the convergence rate of JoBS? What happens if we applied JoBS to only data component? To address these questions, we ran ablations with different fine-tuning time $B_{small}$, training components and plot the best configuration performance at each BO iteration. We used Llama-3-8B-Instruct and the CommonsenseQA evaluation task throughout our ablations.

**Effect of performance scaling law predictor $\mathcal{F}$.** Fig. 4a shows the convergence of JoBS with and without our performance predictor $\mathcal{F}$, given same compute budget. We found that with our performance predictor $\mathcal{F}$ (Sec. 4.2), JoBS (green) initially has a slightly slower convergence rate. This is expected: our observations are noisier at each iteration, causing us to initially learn a noisier

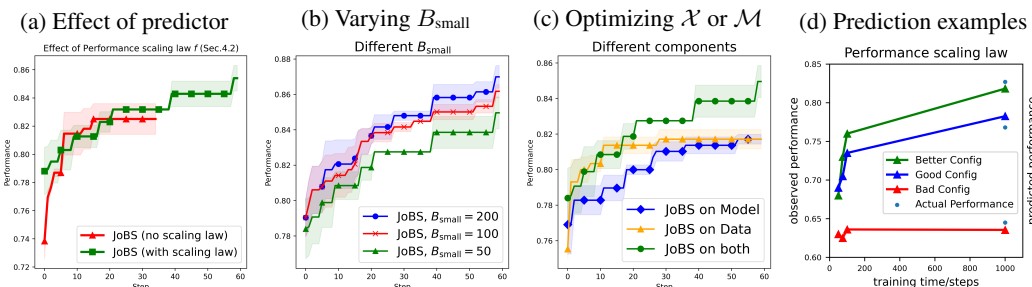

Figure 4: Various ablation studies for the effect of the performance predictor on $\texttt{JoBS}$.

performance landscape. However, using performance scaling laws in $\texttt{JoBS}$ incurs less training time at each iteration (20 times smaller), and thus we can effectively run more BO iterations in total. This enables us to find better training configurations given the same amount of total compute.

**Effect of varying $B_{\text{small}}$.** Fig. 4b illustrates how the choice of fine-tuning time $B_{\text{small}}$ influences the effectiveness of $\texttt{JoBS}$ given a fixed number of BO iterations. As $B_{\text{small}}$ (in seconds) increases, $\texttt{JoBS}$ converges to the better-performing training configurations more quickly. This corroborates our theoretical findings from Theorem 4.1, where a larger $B_{\text{small}}$ means that the observation noise $\epsilon$ associated with our neural network predictor $\mathcal{F}$ is smaller, allowing $\texttt{JoBS}$ to converge more quickly with smaller cumulative regret.

**Effect of varying training components.** Fig. 4c demonstrates the importance of considering both data and model components in $\texttt{JoBS}$. Specifically, applying $\texttt{JoBS}$ (green) to *both* data and model attains much better performance than merely optimizing one of them. We also found that at small number of iterations, optimizing data configurations (blue) produces better results than optimizing model configurations (orange) before converging to similar performances. This suggests that training data mixture plays a larger role than model configurations in improving LLM performance. However, co-optimizing both gives the best results.

**Predicting performance scaling laws.** In Fig.4d, we examined how our performance predictor $\mathcal{F}$ (Sec.4.2) estimates LLM performance under different training configurations. The leftmost points correspond to the true, observed performance at a small training budget, while the rightmost points represent predicted performance after 1000 seconds of training. Of particular note is that good configurations (blue, green) exhibit fruitful scaling laws, with much better performance as training time increases. In contrast, weak configurations (red) are predicted to stagnate, showing little to no gain even with extended training. This shows that our performance scaling law predictor can predict scaling laws dynamically with respect to different configurations selected by $\texttt{JoBS}$. Furthermore, because scaling behavior is highly configuration-dependent, this cannot be captured by a single universal formula found in prior scaling law works.

**Computational cost and other qualitative discussion.** Lastly, we found that $\texttt{JoBS}$ has a smaller runtime than existing baselines, running around 70% to 1240% faster different baselines. We provide a computation cost analysis in App. D, where we find that our performance scaling law predictor is the main reason why $\texttt{JoBS}$ has a smaller runtime, and existing data selection methods are generally expensive. We also present a few interesting analysis of the optimal training configurations found by $\texttt{JoBS}$ in App. E as compared to other baselines.

## 6 CONCLUSION

We illustrated the chicken-and-egg dilemma in LLMs, showing that the interdependence between data and model components makes it challenging for conventional methods to optimize model performance efficiently. We introduced $\texttt{JoBS}$, an efficient algorithm that leverages BO and a novel performance scaling law predictor to jointly optimize data and model configurations by efficiently learning the LLM performance landscape under the fine-tuning regime. Despite noisy estimates from the predictor, $\texttt{JoBS}$ still assures theoretical guarantees and shows promising empirical results in our experiments. Across different evaluation tasks and LLM models, $\texttt{JoBS}$ attains substantial "interaction improvement" over prior baselines, showing that jointly optimizing data and model configurations performs better than independent optimization. We believe $\texttt{JoBS}$ can also be adapted for LLM pretraining, where the same chicken-and-egg dilemma exists.

## 7 ETHICS STATEMENT

Our work strives to improve the performance of LLMs for the greater good. We do not foresee any ethical concerns related to our work. From our theoretical findings and experiments, our method does indeed improve the performance of LLMs.

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

# A  THEORETICAL INSIGHTS INTO OPTIMAL DATA AND TRAINING CONFIGURATIONS

We provide theoretical insights on why an optimal model size and training data size exists in our problem setting. To do so, we analyze the convergence of mini-batch Stochastic Gradient Descent (SGD) (Garrigos & Gower, 2023) over a convex loss function w.r.t. varying model size and training data size (viewed as batch size in this setting). We first present the well-known results on the convergence of the loss function under the mini-batch SGD setting in the proposition below.

**Proposition A.1.** *Let $b$ be the training data size (out of a larger full training dataset) and $m$ be the number of model parameters. Let $T$ be the training steps budget allocated for our model with parameters $\theta_m$. Assume $\theta_m^*$ are the optimal model parameters for the full training dataset and let $f(\theta, \mathcal{X})$ be a convex loss function with respect to model parameters $\theta$ and input examples $\mathcal{X}$. Define the gradient noise as $\sigma_f^* \triangleq Var[\nabla f(\theta_m^*, x)]$ for a randomly sampled datapoint $x$ from the full training data set. Let $L_m$ be the lipschitz constant of the loss function $f$ of a model with $m$ parameters. Lastly, assume $||\theta_m - \theta_m^*||^2 \leq K$ for some constant $K$ and any $m$.*

*If we perform minibatch stochastic gradient descent on $f$ with a randomly sampled data batch of size $b$ (from the full training dataset) on model parameters $\theta_m$ with constant step size $\frac{1}{4L_m}$ for $T$ iterations, then*

$$\mathbb{E}[f(\theta_m^T) - f(_m\theta^*)] \leq \frac{4L_m K}{\sqrt{T}} + \frac{2(n-b)\sigma_f^*}{4L_m b(n-1)\sqrt{T}}, \tag{4}$$

*where $\theta_m^T$ is the model parameters after $T$ SGD steps.*

The above proposition tells us that if the training data is sampled randomly from the full training dataset, the deviation between the optimal loss (over the full training dataset) and the loss w.r.t. learnt parameters $\theta_m^T$ is upper-bounded by the right term in Eq. 4. We can see that the loss w.r.t. learnt parameters $\theta_m^T$ will eventually converge to the optimal loss $f(\theta_m^*)$ w.r.t. increasing training steps.

Interestingly, we observe that the choice of $m, b$, and $T$ is constrained by the given training time budget. We make two assumptions about the relationship between $m, b$ and $T$.

1. **Assumption 1** The lipschitz constant of loss function is governed by $L_m = \frac{c_1}{m}$ for some positive constant $c_1$. This implies the larger the model size, the smaller the lipschitz constant of the loss function (and faster the model learns).

2. **Assumption 2** $T = \frac{bm}{c_2}$ for some positive constant $c_2$. This implies that model and training data size both increases the training budget required to train the model. Given a fixed $T$, we cannot choose a large model size $m$ and training data size $b$.

In the next Theorem, we show that given a training budget $T$, there exists a particular model size $m$ and training size $b$ that will minimize the upper bound in Eq. 4. We would like to emphasize even though a particular choice of $m, b$ could lead to smaller upper bound, it does not necessarily guarantee that the actual deviation in Eq. 4 is smaller (since we are only comparing the upper bounds). However, our theorem provides theoretical insights to possibly explain why training with certain choices of $b, m$ yields better model performance. We have also provided empirical evidence to show that certain training component configurations can produce better-performing LLMs in Sec. 3.

**Theorem A.2.** *Under the same setting as Proposition A.1 and given **Assumption 1** and **2**, for a given training budget of $T$, the upper bound from Proposition A.1 is minimized by solving the following constrained optimization problem:*

$$\min_{m,b} \frac{4c_1 K}{m} + \frac{2(n-b)m\sigma_f^*}{4c_1 b(n-1)} \tag{5}$$
$$s.t. \quad mb = c_2 T$$

*Therefore, an optimal $m, b$ would minimize the constrained optimization problem and minimize the error bounds in Eq. 4.*

## B  PROOF OF THEOREM 3.1

**Theorem 4.1.** *Let $\mathcal{L}(\theta_{\mathcal{X},\mathcal{M},B})$ be the performance landscape of training configuration with bounded RKHS norm: $\|\mathcal{L}\|_\kappa = \sqrt{\langle \mathcal{L}, \mathcal{L} \rangle_\kappa} \leq B$ w.r.t. kernel $\kappa$. Also, let $\gamma_T$ be the maximum information gain from $T$ iterations. As mentioned above, assume we make noisy observation $\hat{\mathcal{L}}(\theta_{\mathcal{X},\mathcal{M},B}) = \mathcal{L}(\theta_{\mathcal{X},\mathcal{M},B}) + \epsilon$ at each BO iteration and error $\epsilon$ associated with our scaling law prediction is Sub-Gaussian with a factor of $R$. Then, running our BO algorithm over training configurations $\mathcal{X}, \mathcal{M}$ with the IGP-UCB acquisition function (Chowdhury & Gopalan, 2017) yields the following cumulative regret with probability at least $1 - \delta$:*

$$R_T = \mathcal{O}\left( B\sqrt{T\gamma_T} + R\sqrt{T}\sqrt{\gamma_T^2 + \gamma_T \ln(1/\delta)} \right) \tag{3}$$

*Proof.* Our proof is divided into two parts. First, we connect our LLM scaling law prediction (Sec. 4.2) to our BO framework and show it can be viewed as *observation noise* $\epsilon$ at each iteration. Then, we show how our scaling law prediction error influences our algorithm by analyzing its cumulative regret with well known results from prior BO works Chowdhury & Gopalan (2017); Srinivas et al. (2010).

To begin, recall that we are trying to maximize our LLM performance, a black-box function $\mathcal{L}(\theta_{\mathcal{X},\mathcal{M},\mathcal{R},B})$ (Sec. 2). Using our scaling law prediction (Sec. 4.2), we instead train our LLM for $B_{\text{small}}$ training steps (or time) and observe $\mathcal{L}(\theta_{\mathcal{X},\mathcal{M},\mathcal{R},B_{\text{small}}})$. We then apply scaling law prediction to observe $\hat{\mathcal{L}}(\theta_{\mathcal{X},\mathcal{M},\mathcal{R},B}) = \mathcal{F}(\mathcal{L}(\theta_{\mathcal{X},\mathcal{M},\mathcal{R},B_{\text{small}}}))$ to estimate what the LLM would have performed if we trained it for the full training duration. Since we are predicting the LLM performance, our model prediction is noisy, with $\mathcal{F}(\mathcal{L}(\theta_{\mathcal{X},\mathcal{M},\mathcal{R},B_{\text{small}}})) = \mathcal{L}(\theta_{\mathcal{X},\mathcal{M},\mathcal{R},B}) + \epsilon$. Hence, we only have access to a noisy estimate of our black-box function: $\mathcal{L}(\theta_{\mathcal{X},\mathcal{M},\mathcal{R},B}) + \epsilon$. Since our estimation error is based on LLM performance, which is bounded (e.g., accuracy), then error $\epsilon \in [0, \alpha]$ with positive constant $\alpha$, and it follows that $\epsilon$ is Sub-Gaussian with a factor $R = \frac{\alpha^2}{4}$ (Arbel et al., 2019).

Hence, we have shown that in our setting, we are making noisy observation of our LLM performance: $\mathcal{L}(\theta_{\mathcal{X},\mathcal{M},\mathcal{R},B}) + \epsilon$ with a Sub-Gaussian error $\epsilon$. This $\epsilon$ is empirically not large (see Fig. 3). Next, we will prove the cumulative regret of our algorithm w.r.t. this observation error. To begin, we present the following lemma from (Chowdhury & Gopalan, 2017)

**Lemma B.1.** *Let $\|f\|_\kappa = \sqrt{\langle f, f \rangle_\kappa} \leq B$. Also, assume that the observation noise associated with each BO iteration is $R$-sub-Gaussian with $R > 0$. Then with probability at least $1 - \delta$, the following holds for BO iteration $t \leq T$:*

$$|\mu_t(x) - f(x)| \leq \left( B + R\sqrt{2(\gamma_t + 1 + \ln(1/\delta))} \right) \sigma_t(x) \tag{6}$$

*where $\gamma_t$ is the maximum information gain after $t$ observations and $\mu_t(x), \sigma_t^2(x)$ are mean and variance of posteror distribution of GP defined in Equation 2, with $\lambda = 1 + 2/T$.*

In our setting, set $f = \mathcal{L}$ (our LLM performance after fine-tuning) and $x = \mathcal{X}, \mathcal{M}, \mathcal{R}$ (our training configuration). This lemma indicates that our estimated mean $\mu_t(x)$ of our performance landscape from our fitted GP over historical observations of LLM performance deviates from the true LLM performance $f(x) = \mathcal{L}(\theta_{\mathcal{X},\mathcal{M},\mathcal{R},B})$ by at most the term in (7).

We are now ready to prove Theorem 4.1. First, we observe that the next training configuration $x_t$ at each BO iteration $t$ is chosen via the IGP-UCB acquisition function (i.e., $x_t = \text{argmax}_x \mu_{t-1}(x) + \beta_t \sigma_{t-1}(x)$ and $\beta_t = B + R\sqrt{2(\gamma_{t-1} + 1 + \ln(1/\delta))}$ where the observation noise associated with each BO iteration is $R$-sub Gaussian). Thus, we can see that at each iteration $t \geq 1$, we have $\mu_{t-1}(x_t) + \beta_t \sigma_{t-1}(x_t) \geq \mu_{t-1}(x^*) + \beta_t \sigma_{t-1}(x^*)$. It then follows that for all $t \geq 1$ and with probability at least $1 - \delta$,

$$
\begin{aligned}
|f(x^*) - f(x_t)| &\overset{(1)}{\leq} \beta_t \sigma_{t-1}(x_t) + \mu_{t-1}(x_t) - f(x_t) \\
&\overset{(2)}{\leq} \beta_t \sigma_{t-1}(x_t) + \mu_{t-1}(x_t) + (\beta_t \sigma_{t-1}(x_t) - \mu_{t-1}(x_t)) \\
&\leq 2\beta_t \sigma_{t-1}(x_t)
\end{aligned}
\tag{7}
$$

where $\overset{(1)}{\leq}$ uses the fact that via Lemma B.1 and our acquisition function, $f(x^*) \leq \beta_t \sigma_{t-1}(x^*) + \mu_{t-1}(x^*) \leq \beta_t \sigma_{t-1}(x_t) + \mu_{t-1}(x_t)$ and $\overset{(2)}{\leq}$ once again uses Lemma B.1.

Using result from Eq. 7, we see that the cumulative regret

$$\sum_{t=1}^{T} r_t = \sum_{t=1}^{T} (f(x^*) - f(x_t)) \leq 2 \sum_{t=1}^{T} \beta_t \sigma_{t-1}(x_t). \tag{8}$$

Since we know that $\sum_{t=1}^{T} \sigma_{t-1}(x_t) = \mathcal{O}(\sqrt{T\gamma_T})$ and used $\beta_t = B + R\sqrt{2(\gamma_{t-1} + 1 + \ln(1/\delta))}$, the cumulative regret in Theorem 4.1 can be written as:

$$R_T = \sum_{t=1}^{T} r_t \tag{9}$$

$$\leq 2 \sum_{t=1}^{T} \beta_t \sigma_{t-1}(x_t) \tag{10}$$

$$\leq 2\mathcal{O}(\sqrt{T\gamma_T})(B + R\sqrt{2(\gamma_{t-1} + 1 + \ln(1/\delta))}) \tag{11}$$

$$= \mathcal{O}\left( B\sqrt{T\gamma_T} + R\sqrt{T}\sqrt{\gamma_T^2 + \gamma_T \ln(1/\delta)} \right). \tag{12}$$

$\square$

## C   MORE EXPERIMENTAL DETAILS

Here, we provide details of how we ran our experiments for JoBS. Our data configuration consists of 10 parameters representing the mixing ratio (a probability simplex) across 10 parameters. Our model configuration consists of 10 parameters, representing:

1. LoRA rank $\in [1, 256]$.
2. Number of LLM layers to apply LoRA to $\in [1, 31]$ (this varies for different LLMs, depending on how many transformer layers are present).
3. Whether to apply LoRA to front layers or rear layers (binary decision).
4. Whether to apply LoRA to Q-projection layer (binary decision).
5. Whether to apply LoRA to V-projection layer (binary decision).
6. Whether to apply LoRA to K-projection layer (binary decision).
7. Whether to apply LoRA to MLP-Up-projection layer (binary decision).
8. Whether to apply LoRA to MLP-Down-projection layer (binary decision).
9. LoRA dropout $\in [0, 1]$.
10. LoRA alpha $\in [1, 500]$.

In all our main results (Table 1, 2, 3), we used 8-shot prompting with CoT. We used 100 BO iterations, with a shortened training time of $B_{small} = 50$ seconds at each iteration. To build our performance scaling law predictor (Sec. 4.2), we collected a random Sobol sequence (Nguyen et al., 2018) of 30 training configurations, their partial and full fine-tuning performance, before training a densely-connected, 64-width, 3 layers neural network $\mathcal{F}$ to predict the full performance. This random sequence is also added to our initial GP model to warm-start BO in JoBS. We used a deep kernel for the GP used to model our LLM performance landscape, and ran our experiments with the Botorch library. At the end of every iteration, we use maximum-likelihood to estimate the hyperparameters in the deep kernel. We normalize and rescale all our training configuration parameters to be between 0

and 1 when fitting our GP. For binary or integer decisions, we use continuous relaxation (Daulton et al., 2022) to project them into the same continuous space as other variables.

Throughout our experiments, we used a single `L40` GPU to fine-tune our LLM.

## D  COMPUTATION COST OF JOBS VERSUS OTHER BASELINES

**Qualitative comparison.** We can actually concisely summarize the computation cost of `JoBS`. We used 30 observations from fully fine-tuning an LLM with random training configurations for 1000 seconds (to learn our performance scaling law predictor and forming the first 30 observations of our trials). Then, we run `JoBS` for 70 iterations, each taking 50 seconds of fine-tuning time. This means `JoBS` uses 33500 seconds (9+ hours) of fine-tuning time. This is faster or comparable to many state-of-the-art data selection algorithms (See next section for a more precise quantitative comparison). For instance, computing the Influence Function (IF) scores (Koh & Liang, 2020) of all data points took a few days. In addition, `JoBS` is an *anytime* algorithm, meaning if computation cost is an issue, we can terminate it at any step to obtain a sub-optimal (but still reasonable good) solution.

**Quantitative comparison of wall-clock hours**

All model selection methods (Liu et al., 2019; He et al., 2024) used in our paper are iterative in nature and require repeated fine-tuning of LLMs. We ensured they run for 33500 seconds. Hence, they have equal computation time (`JoBS` achieves better performance, as seen in Table 3). For data selection methods (LESS, DoReMi, IF, Diversity), we recorded their wall-clock runtime in Table 4. In general, we found data selection methods to be very computationally expensive, taking as much or more time than `JoBS`. One of the key reason that `JoBS` runs faster is due to our scaling law predictor (Sec. 4.2), which drastically reduces the fine-tuning time needed at each BO iteration.

Table 4: Wall -clock runtime comparison of data selection techniques versus `JoBS`

| Method | Time (hours) |
|--------|--------------|
| LESS | 16.3 |
| DoReMi | 18.5 |
| IF | 52 |
| Diversity | 122 |
| `JoBS` | **9.3** |

## E  QUALITATIVE COMPARISON OPTIMAL TRAINING CONFIGURATIONS FOUND BY JOBS VERSUS OTHER BASELINES

Here, we display some of the optimal training configurations found by `JoBS` as compared to other baselines. We divided the configurations into two tables detailing the best data (Table 5) and model (Table 6) configurations found for the **gsm8k** evaluation task. Note that the training domain does not contain **gsm8k** because all our evaluation is done in a much harder out-of-domain setting.

Table 5: Optimal data mixing ratio found by `JoBS` versus other baselines. The columns denote the ratio allocated to each training domain.

| | CQA | HQA | PQA | SciQ | TrivQA | TruthQA | Wiki | MMLU | ARC |
|--------|------|------|------|------|--------|---------|------|------|-----|
| `JoBS` | 0.12 | 0 | 0 | 0.10 | 0.19 | 0 | 0.28 | 0.31 | 0 |
| DoReMi | 0.08 | 0.11 | 0.18 | 0.05 | 0.08 | 0.14 | 0.04 | 0.16 | 0.13 |

On particular interest is that `JoBS` optimizes the data mixture by placing more weights into some data domains based on the evaluation performance on the downstream task (in this case, gsm8k). Specifically, `JoBS` successfully inferred (without knowing that the evaluation task is gsm8k) that domains such as SciQ, TriviaQA, Wikipedia and MMLU contains some math information, and thus chooses them in the optimized data mixture.

On the other hand, DoReMi is a distributionally robust data mixing approach, and results in a more uniform data mixing ratio. This means the data mixture is not tailored specifically to the evaluation task gsm8k, and hence does not perform as well.

Table 6: Optimal model configuration found by JOBS versus other baselines.

|  | Rank | NumLayers | Order | Q | K | V | Up | Down | dropout | $\alpha$ |
|---|---|---|---|---|---|---|---|---|---|---|
| JOBS | 36 | 25 | 1 | 1 | 0 | 1 | 1 | 0 | 0.112 | 64 |
| DARTS | 12 | 13 | 0 | 1 | 1 | 1 | 1 | 0 | 0.058 | 45 |

Next, we examine the optimal model configurations found JOBS. We noticed that JOBS prefers a higher LoRA rank and layer (i.e., how many layer to apply LoRA) but chooses to apply LoRA to only certain transformer layers. In particular, JOBS found that for the gsm8k evaluation task, fine-tuning Q, V, Up layers is sufficient to achieve good fine-tuning performance, and we should fine-tune the rear layers instead of the front layers (Order = 1).

## F   MORE EXPERIMENTAL RESULTS AND DISCUSSION

In Table. 7, 8, 9, 10, 11, we repeated the experimental set-up as those in Table. 1 and mixed and matched different model and data selection methods over another 5 evaluation tasks (CommonsenseQA, HeadQA, MMLU, ARC and TriviaQA). The results show that JOBS outperforms all combinations of data and model selection works. This suggests that jointly adjusting both data and model configurations does indeed produce *interaction improvement* over optimizing the configurations independently. In addition, from running our experiments, we find our approach significantly easier to implement in code.

### F.1   OTHER NAIVE BASELINES

In Table. 12, we jointly optimized training configurations using several other naive approaches in our experiments. We tried 3 naive approaches: (1) **Random** randomly picking 100 different training configurations, fine-tune them for 50 seconds each, use our performance scaling law predictor to predict and select the best-performing training configuration. (2) **Random Data** perform JOBS on model configurations for only 10 iterations and repeat the experiment with 10 randomly chosen data configurations (this ensures the same amount of compute as performing JOBS on all training configurations for 100 iterations). (3) **Random Model** repeat approach (2) on training configurations instead. While these approaches serve as good sanity checks, they do not yield good LLM performances, largely because randomly selecting training configurations does not exploit the learnt performance landscape from historically observed LLM performances.

Table 7: CommonsenseQA (Talmor et al., 2019)

| ↓ Model \| Data → | Default | LESS | DoReMi | IF | Diversity | BO | JOBS |
|---|---|---|---|---|---|---|---|
| Default | $76.3_{\pm 1.0}$ | $73.0_{\pm 0.8}$ | $74.2_{\pm 1.7}$ | $79.3_{\pm 0.7}$ | $77.4_{\pm 1.7}$ | $80.6_{\pm 0.8}$ | - |
| DARTS | $79.6_{\pm 1.3}$ | $76.3_{\pm 1.7}$ | $76.1_{\pm 1.1}$ | $73.7_{\pm 1.2}$ | $80.1_{\pm 1.1}$ | $79.6_{\pm 0.6}$ | - |
| AutoLoRA | $78.9_{\pm 0.9}$ | $79.8_{\pm 0.4}$ | $76.1_{\pm 0.5}$ | $77.9_{\pm 1.2}$ | $78.0_{\pm 1.0}$ | $81.5_{\pm 1.0}$ | - |
| RoBoT | $74.9_{\pm 0.8}$ | $75.5_{\pm 0.9}$ | $77.1_{\pm 0.9}$ | $79.4_{\pm 1.5}$ | $76.3_{\pm 0.9}$ | $80.2_{\pm 0.2}$ | - |
| BO | $79.7_{\pm 1.3}$ | $79.4_{\pm 0.3}$ | $77.0_{\pm 0.4}$ | $81.1_{\pm 0.9}$ | $79.4_{\pm 1.1}$ | $80.7_{\pm 1.2}$ | - |
| JOBS | - | - | - | - | - | - | $\mathbf{84.3}_{\pm 2.4}$ |

Table 8: HeadQA (Vilares & Gómez-Rodríguez, 2019)

| ↓ Model \| Data → | Default | LESS | DoReMi | IF | Diversity | BO | JoBS |
|---|---|---|---|---|---|---|---|
| Default | $47.0_{\pm0.9}$ | $46.4_{\pm1.0}$ | $46.3_{\pm0.8}$ | $46.1_{\pm0.7}$ | $45.8_{\pm1.2}$ | $49.2_{\pm0.6}$ | - |
| DARTS | $43.6_{\pm0.2}$ | $46.7_{\pm1.3}$ | $53.0_{\pm2.4}$ | $40.7_{\pm1.5}$ | $47.3_{\pm0.7}$ | $48.9_{\pm1.2}$ | - |
| AutoLoRA | $49.1_{\pm1.4}$ | $49.4_{\pm0.4}$ | $50.3_{\pm0.9}$ | $47.7_{\pm1.1}$ | $48.4_{\pm1.0}$ | $51.3_{\pm0.3}$ | - |
| RoBoT | $49.5_{\pm1.2}$ | $48.0_{\pm1.0}$ | $48.7_{\pm1.7}$ | $49.2_{\pm1.2}$ | $50.6_{\pm0.8}$ | $50.8_{\pm0.6}$ | - |
| BO | $49.6_{\pm0.8}$ | $51.3_{\pm1.0}$ | $52.0_{\pm0.6}$ | $52.6_{\pm0.3}$ | $50.3_{\pm0.7}$ | $48.2_{\pm0.6}$ | - |
| JoBS | - | - | - | - | - | - | $\underline{\mathbf{55.8}}\pm1.5$ |

Table 9: MMLU (Hendrycks et al., 2021)

| ↓ Model \| Data → | Default | LESS | DoReMi | IF | Diversity | BO | JoBS |
|---|---|---|---|---|---|---|---|
| Default | $61.2_{\pm1.3}$ | $63.5_{\pm0.9}$ | $59.7_{\pm1.8}$ | $57.9_{\pm0.6}$ | $62.1_{\pm1.4}$ | $64.2_{\pm1.2}$ | - |
| DARTS | $58.3_{\pm0.7}$ | $61.0_{\pm2.1}$ | $62.9_{\pm1.0}$ | $55.7_{\pm1.6}$ | $60.1_{\pm0.5}$ | $63.4_{\pm2.0}$ | - |
| AutoLoRA | $62.5_{\pm1.4}$ | $64.3_{\pm0.6}$ | $60.8_{\pm2.2}$ | $58.2_{\pm1.9}$ | $63.7_{\pm0.8}$ | $61.5_{\pm1.1}$ | - |
| RoBoT | $59.9_{\pm0.9}$ | $60.7_{\pm1.2}$ | $63.4_{\pm1.7}$ | $61.5_{\pm1.5}$ | $58.3_{\pm2.3}$ | $62.1_{\pm0.7}$ | - |
| BO | $55.8_{\pm1.8}$ | $57.2_{\pm0.4}$ | $61.3_{\pm1.2}$ | $60.5_{\pm1.6}$ | $63.9_{\pm1.0}$ | $59.6_{\pm1.5}$ | - |
| JoBS | - | - | - | - | - | - | $\mathbf{69.5}_{\pm0.8}$ |

Table 10: ARC (Clark et al., 2018)

| ↓ Model \| Data → | Default | LESS | DoReMi | IF | Diversity | BO | JoBS |
|---|---|---|---|---|---|---|---|
| Default | $54.7_{\pm1.3}$ | $59.2_{\pm0.7}$ | $61.4_{\pm2.0}$ | $52.8_{\pm1.5}$ | $60.6_{\pm0.9}$ | $62.3_{\pm1.2}$ | - |
| DARTS | $58.1_{\pm0.8}$ | $61.0_{\pm1.6}$ | $62.8_{\pm0.5}$ | $54.3_{\pm2.1}$ | $57.9_{\pm1.7}$ | $60.5_{\pm0.6}$ | - |
| AutoLoRA | $60.4_{\pm1.1}$ | $63.2_{\pm0.9}$ | $58.6_{\pm1.9}$ | $55.1_{\pm0.8}$ | $62.1_{\pm2.0}$ | $59.8_{\pm1.0}$ | - |
| RoBoT | $56.8_{\pm1.5}$ | $58.7_{\pm1.4}$ | $61.1_{\pm1.2}$ | $60.3_{\pm0.7}$ | $55.7_{\pm2.2}$ | $61.4_{\pm1.3}$ | - |
| BO | $52.6_{\pm2.0}$ | $55.9_{\pm0.6}$ | $59.7_{\pm1.3}$ | $58.5_{\pm1.4}$ | $63.4_{\pm1.0}$ | $57.4_{\pm0.9}$ | - |
| JoBS | - | - | - | - | - | - | $\mathbf{70.4}_{\pm1.3}$ |

Table 11: TriviaQA Gen (Joshi et al., 2017)

| ↓ Model \| Data → | Default | LESS | DoReMi | IF | Diversity | BO | JoBS |
|---|---|---|---|---|---|---|---|
| Default | $55.5_{\pm1.4}$ | $57.2_{\pm0.8}$ | $53.1_{\pm0.9}$ | $55.8_{\pm0.7}$ | $58.9_{\pm0.8}$ | $65.0_{\pm0.6}$ | - |
| DARTS | $58.2_{\pm0.8}$ | $61.3_{\pm1.2}$ | $61.0_{\pm0.7}$ | $63.3_{\pm1.0}$ | $59.2_{\pm0.6}$ | $66.7_{\pm1.8}$ | - |
| AutoLoRA | $67.8_{\pm1.4}$ | $64.7_{\pm0.9}$ | $70.6_{\pm2.2}$ | $68.6_{\pm1.7}$ | $66.2_{\pm1.5}$ | $69.7_{\pm2.4}$ | - |
| RoBoT | $58.4_{\pm1.5}$ | $62.3_{\pm1.7}$ | $64.2_{\pm1.4}$ | $57.2_{\pm1.2}$ | $63.4_{\pm1.5}$ | $68.2_{\pm1.3}$ | - |
| BO | $70.7_{\pm1.4}$ | $66.7_{\pm0.8}$ | $72.5_{\pm0.8}$ | $71.7_{\pm0.9}$ | $74.7_{\pm1.0}$ | $72.7_{\pm2.3}$ | - |
| JoBS | - | - | - | - | - | - | $\underline{\mathbf{76.2}}\pm1.9$ |

Table 12: Comparison of some naive baselines with JoBS (Higher is better), averaged over 5 trials. **Random Data** means we randomly selected data mixtures and applied JoBS only on the model configurations (vice versa for **Random Model**). **Random** means we randomly selected training configurations.

| Model | Task | **Random** | **Random Data** | **Random Model** | JoBS |
|---|---|---|---|---|---|
| Llama-3-8B-Instruct | gsm8k | $66.5_{\pm2.4}$ | $67.3_{\pm1.6}$ | $71.5_{\pm0.9}$ | $80.4_{\pm1.9}$ |
| | TruthfulQA | $59.1_{\pm1.9}$ | $59.8_{\pm1.5}$ | $64.2_{\pm1.4}$ | $75.8_{\pm1.3}$ |
| | CommonsenseQA | $78.8_{\pm3.2}$ | $76.4_{\pm1.2}$ | $76.3_{\pm1.2}$ | $84.3_{\pm2.4}$ |
| | HeadQA | $51.5_{\pm2.1}$ | $51.3_{\pm2.1}$ | $53.2_{\pm1.2}$ | $55.8_{\pm1.5}$ |
| | MMLU | $67.6_{\pm2.9}$ | $66.4_{\pm0.7}$ | $63.1_{\pm1.1}$ | $69.5_{\pm0.8}$ |
| | ARC | $60.5_{\pm3.2}$ | $65.2_{\pm1.7}$ | $64.6_{\pm0.6}$ | $70.4_{\pm1.3}$ |
| | TriviaQA | $58.2_{\pm3.6}$ | $61.7_{\pm2.4}$ | $63.2_{\pm1.5}$ | $76.2_{\pm1.2}$ |

# G    ADDITIONAL ABLATIONS AND DISCUSSION

This section highlights the additional experimental results and discussions we run during the rebuttal, specifically on the CommonsenseQA task. To summarize,

1. First, in Fig. 5, we run ablations on the predictor predictor validation set error for LLM model of different sizes (Llama-3-8B-Instruct, Qwen3-14B and Qwen3-32B). Since our task performance is more than 80%, the predictor error is reasonable across different model sizes. In addition, JoBS's BO backbone handles these prediction error gracefully as observation noise and our algorithm still converges and performs better than other baseline (Theorem 4.1).

2. Second, in Fig. 6, we run ablations on how the number of training samples influence our predictor $\mathcal{F}$'s prediction error. In general, the results show that more training samples allow our predictor to be more accurate. However, fitting the predictor with more training samples is more computationally expensive since using more training samples reduces the number of BO function evaluations. From our experiments, using 30 training samples is sufficient to yield good LLM performance.

3. Third, we ran additional experiments in Table 13 to showcase the effectiveness of JoBS on LoRA fine-tuning of larger models (averaged over 5 trials).

4. Fourth, we ran additional experiments to investigate how different number of training samples influence JoBS's performance downstream in Table 14 (averaged over 5 trials).

5. Fifth, we ran additional experiments to investigate JoBS's performance for full-parameter fine-tuning in Table 15. The model configuration we optimize is a one-dimensional variable indicating the number of layers in which we apply the full-parameter fine-tuning to (averaged over 5 trials).

## G.1    ABLATION STUDY ON PREDICTION ERROR OF NEURAL NETWORK PREDICTOR $\mathcal{F}$

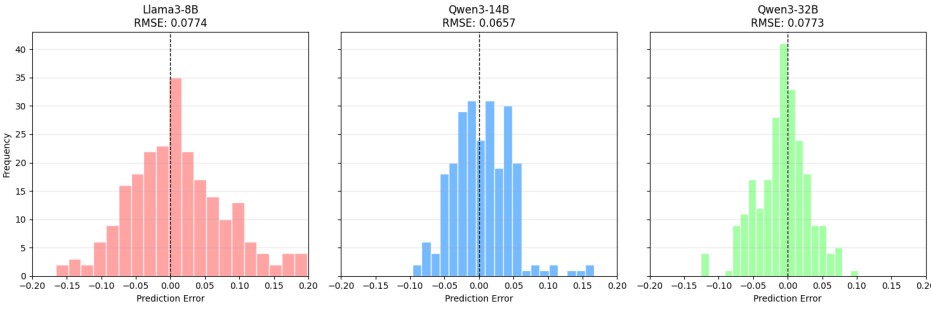

Figure 5: Predictor error (on validation set) across varying model sizes. Predictor learnt from performance observations of larger models. This hints that performance of larger models is easier to extrapolate, possibly due to more stable training dynamics.

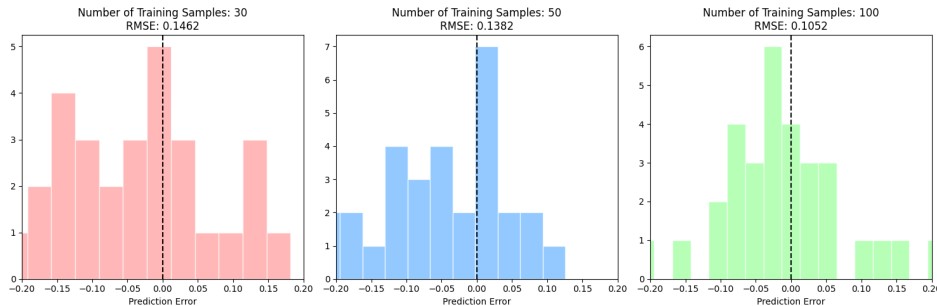

Figure 6: Predictor error (on validation set) for varying number of training samples, with Llama-3-8B-Instruct.

Table 13: JoBS applied to LoRA for PEFT of larger models.

| ↓ Model | Method → | LESS + AutoLoRA | DoReMi + DARTS | JoBS |
|---|---|---|---|
| Llama-3-8B-Instruct | 0.80 | 0.79 | **0.84** |
| Qwen3-14B | 0.82 | 0.80 | **0.86** |
| Qwen3-32B | 0.83 | 0.84 | **0.90** |

Table 14: Performance of JoBS w.r.t. different number of samples used to train $\mathcal{F}$ (Sec. 4.2)

| ↓ Task | Training samples → | 30 | 100 |
|---|---|---|
| CommonsenseQA | 0.84 | **0.88** |

Table 15: JoBS applied to full-parameter fine-tuning of larger models.

| ↓ Model | Method → | LESS + AutoLoRA | DoReMi + DARTS | JoBS |
|---|---|---|---|
| Llama-3-8B-Instruct | 0.73 | 0.76 | **0.81** |
| Qwen3-14B | 0.76 | 0.81 | **0.83** |
| Qwen3-32B | 0.86 | 0.82 | **0.88** |

## G.2 RELATED WORK ON SCALING LAW PREDICTORS & BO JOINT-OPTIMIZATION

**Scaling Law Predictors** Understanding how LLM performance scales with training resources is crucial for efficient optimization. Foundational works have established power laws relating loss to model size, dataset size, and compute budget (Kaplan et al., 2020; Hoffmann et al., 2022; Zhang et al., 2024a; Shukor et al., 2025). More recent studies have extended these laws to predict downstream performance on specific metrics (Wu & Tang, 2024; Chen et al., 2025b) and optimize data mixtures (Chen et al., 2025c; Xie et al., 2023a; Ye et al., 2024) . However, these approaches typically derive static formulas by assuming fixed model architectures or training recipes. Unlike these rigid scaling laws, JoBS employs a flexible neural predictor capable of estimating performance across a diverse, dynamic search space of joint data and model configurations, enabling the evaluation of "interaction improvements" without exhaustive full-scale training.

**Bayesian Optimization (BO)** BO has been widely adopted for optimizing black-box functions where evaluations are costly (Srinivas et al., 2010). In the context of deep learning, BO has been successfully applied to Neural Architecture Search (NAS) (White et al., 2020) and hyperparameter tuning (Brochu et al., 2010; Snoek et al., 2012). To handle the complexity of modern training setups, recent works have explored methods such as introducing constrained BO for resource management (Eriksson & Poloczek, 2021) and mixed-variable optimization for combinations of discrete and continuous parameters (Daulton et al., 2022). Frameworks like AutoAI (Chen et al., 2024) have also attempted to optimize general machine learning pipelines, they do not specifically address the "chicken-and-egg" interdependency between data mixtures and PEFT configurations in LLMs. JoBS leverages these advanced BO techniques —specifically deep kernel learning (Wilson et al., 2016) —to navigate this complex, high-dimensional landscape efficiently.

## G.3 ADDITIONAL EXPERIMENTAL RESULTS ON MULTI-TASK FINE-TUNING

We also ran JoBS on a multi-task scenario, where one trains the predictor and applies JoBS such that the LLM will perform well across multiple tasks at once. In the multi-task scenario, we average the LLM performance over 5 different evaluation tasks: TruthfulQA, TriviaQA, CommonsenseQA, GSM8K, and MMLU, and treat this average performance as our optimization objective.

Table 16: Comparison of different data mixing methods across model sizes for the multi-task scenario.

| Model | LESS + AutoLoRA | DoReMi + DARTS | JoBS with multi-task predictor |
|---|---|---|---|
| Llama-3-8B-Instruct | 0.63 | 0.66 | **0.70** |
| Qwen3-14B | 0.71 | 0.66 | **0.73** |
| Qwen3-32B | 0.74 | 0.72 | **0.79** |

