# OpenReview forum: "The Chicken and Egg Dilemma: Co-optimizing Data and Model Configurations for LLMs"
_ICLR.cc/2026/Conference — Submitted to ICLR 2026_

### Official Review · Reviewer_W1bf · 2025-10-21

**Soundness:** 2
**Presentation:** 3
**Contribution:** 2
**Rating:** 4
**Confidence:** 3

**Summary:**

This paper tackles the classic “chicken-and-egg” problem in LLM training — the strong interdependence between data configuration and model configuration — and proposes JoBS, a method that jointly optimizes both by modeling the performance landscape with Gaussian processes and accelerating search using a performance scaling predictor. Experiments show that JoBS consistently outperforms independent optimization approaches: for example, it achieves 80.4% accuracy on gsm8k (vs. ~74.7% for the best baseline) and 75.8% on TruthfulQA (vs. ~71.7%), delivering a stable 6–7% interaction improvement across tasks and models (around 7B). Thanks to its predictor, JoBS can forecast the final training performance based on early-stage results, significantly reducing the time cost. It is worth noting that the experiments are conducted under the LoRA setting and do not explore other tunable training parameters.

**Strengths:**

- The predictor proposed in the paper is simple yet effective. Predicting the final training outcome based on early-stage performance is an excellent idea, and the experiments demonstrate that this approach alleviates the problem of reduced time efficiency caused by an excessively large search space.

- The interdependence between data configuration and model configuration is indeed an important aspect of LLM training. The use of Bayesian optimization gives this approach the potential to be extended to different model architectures and various task domains.

- The paper conducts extensive experiments based on the LoRA configuration, and the results demonstrate the reliable performance improvements of JoBS over the baselines, further validated across multiple cross-domain datasets. Additional supplementary experiments and analyses also provide deeper insights.

**Weaknesses:**

- Although the predictor can help forecast results when trained properly, it remains a risky choice: an unusual training curve (e.g., one rarely seen in the training data) might cause a promising training configuration to be overlooked. Since LLM training is influenced by many hyperparameters, predictor failures may occur frequently. Moreover, the need to train a new predictor for each task limits the method’s applicability and stability in multi-task joint SFT scenarios.

- The experiments in the paper seem to be limited to the LoRA SFT setting. While LoRA has indeed become the primary choice for downstream fine-tuning under limited compute, considering that hyperparameter space search itself already requires substantial time and resources, it may be necessary to demonstrate JoBS’s performance on full-parameter fine-tuning (which could also introduce new challenges for components such as the predictor). This would make JoBS more practically valuable for real-world applications.

- JoBS also lacks experiments on larger models beyond 7B, such as 32B or 72B. Moreover, considering the current development of LLMs, JoBS performs data configuration optimization based on proportion search within a fixed data pool rather than dynamically constructing or filtering data. This raises concerns about its robustness in “open-domain” or highly variable data distribution scenarios — conditions that are common in real-world applications where training datasets are continuously expanding. As a result, introducing JoBS in such settings could potentially lead to additional training costs.

**Questions:**

- Can the paper provide evidence of the predictor’s performance in multi-task joint SFT scenarios?

- Can the paper provide results of JoBS in more realistic settings, such as full-parameter fine-tuning, larger model scales, or more open-ended tasks?

- Can the authors provide more analysis or evidence on how robust this predictor is to noisy or unstable early training dynamics, which are common in large-scale LLM training (e.g., due to optimizer warm-up, curriculum learning, or data distribution shifts)?

---

> ### Author Response · Authors · 2025-11-24
> **Response Part 1/2**
>
> Thank you for reviewing our work. We appreciate your acknowledgement of our paper's strengths (e.g., "excellent" idea of using the performance predictor and praising our empirical results). We would like to address your comments and hope our clarifications will improve your opinion of our work.
>
> > **Although the predictor can help forecast results when trained properly, it remains a risky choice: an unusual training curve (e.g., one rarely seen in the training data) might cause a promising training configuration to be overlooked.**
>
> We appreciate the reviewer's concerns regarding the potential for unusual training curves. While it is indeed true that we do not see all possible training dynamics (including the unusual training curve rarely seen in the training data) when collecting initial training configuration samples, our neural network predictor, being a flexible function approximator, can generalize well to unseen training configurations, and thus still predict LLM performance well enough to efficiently find optimal training configurations using Bayesian Optimization in JoBS.
>
> In addition, it is worth noting that even when our predictor does not make perfect predictions, our algorithm JoBS is *robust to noisy predictions* (see Theorem 4.1) and provides a principled way to correlate different performance observations to each other in Bayesian Optimization. Hence, the choice of a neural network predictor is well-justified in our setting, allowing us to efficiently assess the quality of a training configuration.
>
>
> > **While LoRA has indeed become the primary choice for downstream fine-tuning under limited compute, it may be necessary to demonstrate JoBS’s performance on full parameter fine-tuning (which could also introduce new challenges for components such as the predictor).**
>
> Thank you for the suggestion. To demonstrate that JoBS can be extended to full-parameter fine-tuning, *we have conducted additional full-parameter fine-tuning experiments on Llama-3-8B-Instruct* for the CommonsenseQA task. In addition, *we also performed full-parameter fine-tuning on the larger Qwen3-14B and Qwen3-32B LLMs* for the same task, to illustrate that JoBS can be extended to larger models. The wide variety of model families and sizes included in our experiments should make our empirical results more convincing. Our results in the table below show that *even for larger models, JoBS still outperforms other baselines with full-parameter fine-tuning*. We have included these additional results in App. G of our revised paper.
>
> | $\downarrow$ Model / Method $\rightarrow$ | LESS + AutoLoRA | DoReMi + DARTS | JoBS |
> | :---: | :---: | :---: | :---: |
> | Llama-3-8B-Instruct | 0.73 | 0.76 | **0.81** |
> | Qwen3-14B | 0.76 | 0.81 | **0.83** |
> | Qwen3-32B | 0.86 | 0.82 | **0.88** |
>
> In addition, we did not empirically observe that full-parameter fine-tuning added significant new challenges for the predictor. As shown in the results above, JoBS still outperforms the baselines in this setting. Furthermore, Fig. 5 in App. G shows that the predictive error is even smaller for larger models. This suggests that scaling up model size leads to more reliable performance scaling, which in turn generates higher-quality training samples for the performance predictor.
>
> > **Can the authors provide more analysis or evidence on how robust this predictor is to noisy or unstable early training dynamics**
>
> We have added more ablations to our revised paper's App. G where we used different number of training samples to fit the predictor. In general, we find that (Fig. 6) using more training samples increases the predictor's accuracy. However, despite using only *30 training samples* to train our predictor, *JoBS still performs better than existing baselines* (Table 3). This empirically demonstrates the robustness of JoBS to predictive errors. Furthermore, we observe that *the predictor better captures the training dynamics of larger LLMs compared to smaller models* (Fig. 5). This suggests that larger models have more consistent performance scaling patterns, hence producing higher-quality training samples for the performance predictor.

---

> ### Author Response · Authors · 2025-11-24
> **Response Part 2/2**
>
> > **Can the paper provide evidence of the predictor’s performance in multi-task joint SFT scenarios?**
>
> We thank the reviewer for their question. To start off, we interpret the reviewer's reference to "multi-task joint SFT scenario" as the setting where one trains the predictor and applies JoBS such that the LLM will perform well across multiple tasks at once. In this case, we can adapt JoBS by making a slight modification to the overall algorithm such that instead of trying to optimize the LLM performance w.r.t. a specific task, we optimize the LLM w.r.t. multiple tasks at once.
>
> To empirically investigate whether our neural network predictor can accurately predict performance in the multi-task scenario, we average the LLM performance over 5 different evaluation tasks: TruthfulQA, TriviaQA, CommonsenseQA, GSM8K, and MMLU, and treat this average performance as our optimization objective.
>
> First, we find that from 30 training samples (collected in the same manner as that in our paper in Sec. 4.2), our predictor's root mean squared error is around 0.102, which is consistent with the predictive errors we observed in our original experiments (shown in App. G in our revised paper). This suggests that our predictor still works reasonably well.
>
> Next, we run JoBS with this newly trained predictor to optimize our training configurations to maximize aggregated LLM performance ("multi-task scenario"). From the table below (Table 16), we observe that *JoBS is still able to optimize the training configurations better than existing baselines*.
>
> | $\downarrow$ Model / Method $\rightarrow$ | LESS + AutoLoRA | DoReMi + DARTS | JoBS with multi-task predictor|
> | :---: | :---: | :---: | :---: |
> | Llama-3-8B-Instruct | 0.63 | 0.66 | **0.70** |
> | Qwen3-14B | 0.71 | 0.66 | **0.73** |
> | Qwen3-32B | 0.74 | 0.72 | **0.79** |
>
> It is worth noting that the performance for "multi-task scenarios" is *generally slightly lower* than a single-task scenario in our paper's original experiments (which are single-task). This is not surprising because it is harder to fine-tune an LLM to perform well over multiple tasks. We hope this evidence has clarified your question regarding our predictor's and JoBS's performance on "multi-task joint SFT scenarios".
>
>
> > **JoBS performs data configuration optimization based on proportion search within a fixed data pool rather than dynamically constructing or filtering data. This raises concerns about its robustness in “open-domain” or highly variable data distribution scenarios**
>
> Thank you for the feedback. We interpret your question as referring to scenarios where the data pool or data domain is constantly changing. Based on this understanding, we offer the following clarifications.
>
> JoBS performs data optimization by selecting a data subset, from an existing pool of data, parameterized by the data mixing ratio. If a certain pool of data is overly noisy or irrelevant for the evaluation task, its ratio will be reduced to zero via the optimization process. This formulation of data mixtures aligns with existing literature on data selection and optimization [1,2]. It is also practical in real-world settings because data can usually be categorized into different groups (i.e, domains) such as different data streams or demographics of users.
>
> In cases where the data distribution scenarios are "highly variable", it is possible that the pool of data points changes continually over time. However, this is not an issue for JoBS. As long as we have well-defined data domains (or categories) and each data point is assigned to a data domain, JoBS can still sample from these data domains (even though the exact data in these domains might change) to fulfil the data mixing ratio proposed by Bayesian Optimization.
>
> Furthermore, for settings where the data domains themselves shift significantly over time, a practical approach is to re-run JoBS at regular intervals (decided by business needs) to obtain new training data mixtures to fine-tune the LLM. This is aligned with common practice for many front-facing LLMs (e.g., ChatGPT, Gemini, DeepSeek), which are repeatedly fine-tuned using newly acquired data. We hope this answers the reviewer's question.
>
> ---
> We sincerely thank the reviewer for their constructive feedback and detailed analysis. All additional experiments and ablation studies have been included in App. G of our revised paper. We hope that these additions and our responses have satisfactorily addressed your concerns and improved your opinion of our work.
>
> ---
>
> [1] Xia et al. LESS: Selecting Influential Data for Targeted Instruction Tuning. 2024
>
> [2] Wang et al. Diversity Measurement and Subset Selection for Instruction Tuning Datasets. 2024

---

> > ### Comment · Reviewer_W1bf · 2025-11-25
> > **Thank you for your rebuttal.**
> >
> > The rebuttal clarified some points of confusion. I have raised the soundness score, but I maintain my original assessment of the overall score.

---

> > > ### Author Response · Authors · 2025-11-25
> > >
> > > Thank you for your follow-up and for updating the soundness score.
> > >
> > > We appreciate the time you spent reviewing our work and the opportunity to clarify points in the rebuttal.
> > >
> > > Since you noted that our rebuttal resolved several points of confusion and increased our soundness score, could you kindly provide us additional guidance on any remaining concerns you might still have?

---

> > > > ### Comment · Reviewer_W1bf · 2025-11-26
> > > > **Thanks you for your rebuttal.**
> > > >
> > > > I do not doubt that the method in the article can be applied to dynamic datasets. However, when the dataset changes, the method must be reapplied to ensure correctness. In large-scale SFT and pretraining scenarios, this cost overhead is worth considering. For 32B or even 72B models, I remain skeptical about whether the performance improvement brought by such dynamic overhead is justified. Therefore, I maintain the original score.

---

> ### Author Response · Authors · 2025-12-02
>
> We thank the reviewer for their continued engagement. We address their remaining concerns below:
> >... when the dataset changes, the method must be reapplied to ensure correctness. In large-scale SFT and pretraining scenarios, this cost overhead is worth considering.
>
> Even in dynamic dataset scenarios, we still need to re-apply existing methods to re-optimize training configurations. JoBS is *significantly more efficient than existing methods*. As shown in Table 4, JoBS requires only 9.3 hours, whereas baselines like LESS (16.3 hours) and Diversity (122 hours) incur significantly higher overheads. Therefore, while there is indeed an overhead to consider, it is significantly lower than existing selection pipelines.
>
> >  For 32B or even 72B models, I remain skeptical about whether the performance improvement brought by such dynamic overhead is justified.
>
> Our additional experiments on larger models (Table 13, App. G) demonstrate that JoBS continues to outperform existing baselines at scale by up to 7\%. In the context of large-scale 32B or 72B model training where computational budgets are massive and even 1% gains are critical, we believe that such a significant improvement actually highly justifies the comparatively minor cost of running JoBS.
>
> Once again, we thank the reviewer for their time and effort in engaging with us. We hope that our clarifications have improved your opinion of our work.

---

### Official Review · Reviewer_AUVd · 2025-10-28

**Soundness:** 3
**Presentation:** 4
**Contribution:** 3
**Rating:** 6
**Confidence:** 3

**Summary:**

The paper addresses the "chicken-and-egg dilemma" in LLM optimization: the ideal training data configuration (e.g., data mixture) depends on the model configuration (e.g., PEFT hyperparameters), and vice versa. Optimizing these factors independently leads to suboptimal results. The authors propose JOBS (Joint Bayesian Optimization with Scaling Laws), a method that frames this as a black-box optimization problem. JOBS uses Bayesian Optimization (BO) to efficiently search the joint configuration space. To make this process computationally feasible, it introduces a neural network-based performance predictor that extrapolates final model performance from short, inexpensive training trials. The authors demonstrate empirically that this joint optimization approach significantly outperforms methods that optimize data and model configurations independently or in an alternating fashion, while also being much faster.

**Strengths:**

The paper designs its method using BO techniques flexibly and appropriately. The core idea of using a performance predictor to amortize the cost of BO evaluations is well-motivated and practical. The writing is easy to follow, and the overall narrative is clear. The empirical results look good and strong, showing consistent improvements over a wide range of baselines, including independent and alternating optimization schemes, across multiple tasks and models. The "interaction improvement" claim is well-supported by the main results tables.

**Weaknesses:**

The problem formulation relies on a fixed training time budget. However, training time is highly sensitive to the implementation (e.g., specific frameworks for PEFT or inference). It is questionable whether using time as the primary budget is a robust choice, as opposed to a more implementation-agnostic budget like total tokens or training steps or FLOPs or other potential choices.

The motivation in Section 3, particularly Figure 2, is a key pillar of the paper. However, I am wondering if the performance variance shown in Figure 2 is due to training instability. Many papers suggest that PEFT can result in high variance in training performance. The paper does not seem to address this; for instance, it's not clear if each point in the landscape is an average of multiple runs. If the variance from training instability has a greater magnitude than the "valleys" created by the model/data strategy itself, then Figure 2 cannot faithfully support the claim that the landscape is smooth but complex.

For Section 2, I don't think this makes for a comprehensive related work review. The paper introduces BO and scaling laws as core components, but the related works section does not sufficiently discuss prior work using BO and scaling laws specifically for training configuration optimization and performance prediction.

I am curious why a neural network was chosen as the scaling law predictor. A scaling law is usually a symbolic expression, which makes it explainable and allows it to generalize to unseen extrapolated settings (as the law is not overly complex). A neural network, while a flexible function approximator, can fit the training data well but may not generalize well, is a black-box predictor, and may exhibit non-smooth behavior. The paper could benefit from a clearer justification for this choice over more traditional scaling law formulations.

(minor) Using $\mathcal L$ as the performance metric notation is weird. This notation almost universally denotes a loss function, so using it for accuracy or another performance metric is confusing.

**Questions:**

1. The paper studies "data mixture" as a form of data strategy, parameterizing the data configuration as a probability simplex over N datasets. However, in many real applications, there are no clear "datasets," and data selection is performed instance-wise. How do you see this method handling such a scenario? For example, can this method be scaled to "instance-level" data selection (e.g., where we have many data points and each is labeled as "selected" or "excluded")? This would dramatically increase the dimensionality of the data configuration space $\mathcal{X}$.
2. The entire method relies on the Gaussian Process (GP) surrogate being a good model of the true performance landscape $\mathcal{L}$. How good can the GP depict the real metric of $\mathcal{L}$? Although it is computationally expensive to validate this exhaustively, can we get a sense of the surrogate model's fidelity from some smaller-scale experiments (e.g., by comparing the GP's predictions to a densely-sampled ground truth in a low-dimensional version of the problem)?

---

> ### Author Response · Authors · 2025-11-24
> **Response Part 1/3**
>
> Thank you for the positive score and encouraging comments on our method's practicality, motivation, and well-supported empirical results. We hope our response will answer your remaining questions and improve your opinion of our work.
>
> > **The problem formulation relies on a fixed training time budget. However, training time is highly sensitive to the implementation (e.g., specific frameworks for PEFT or inference) [...] as opposed to a more implementation-agnostic budget like total tokens or training steps or FLOPs**
>
> Thank you for the comment. The budget $B$ in our problem formulation (Eq. 1) is indeed a generalizable constraint -- the training time budget used in our paper can be replaced by other constraint choices such as total training steps or tokens.
>
> However, we choose to work with a training time budget primarily because it is *"easier for practitioners to interpret"* (Line 115) and GPU-hours (which is directly associated with costs) is one of the most important factors considered by practitioners running real-world experiments. For instance, the budget allocated to a machine learning project in the industry is typically determined by GPU-hours (and associated cost) for LLM fine-tuning. We could consider other budget choices, such as training tokens or training steps, but they often lack a strong correlation with the GPU-hours and monetary constraints encountered in real-world settings. Therefore, incorporating the practical constraint of a training time budget into our problem formulation is not a limitation, but rather a core strength of our work.
>
> In addition, by maintaining a consistent framework (LoRA in PEFT) across all experiments in our paper, we *effectively control for implementation-specific variance* and ensure that our results remain fair and comparable. Crucially, regardless of the chosen implementation framework, practitioners are _still_ limited by training time in practical settings (e.g., GPU-hours), instead of other budget metrics such as training steps or FLOPs.
>
>
> > **I am wondering if the performance variance shown in Figure 2 is due to training instability. Many papers suggest that PEFT can result in high variance in training performance. The paper does not seem to address this; for instance, it's not clear if each point in the landscape is an average of multiple runs. If the variance from training instability has a greater magnitude than the "valleys" created by the model/data strategy itself, then Figure 2 cannot faithfully support the claim that the landscape is smooth but complex.**
>
> We appreciate the reviewer raising this point regarding PEFT's training instability. We would like to clarify that to mitigate this exact issue, each data point in the performance landscape (Fig. 2) represents the *average of 5 independent trials*. We observed a standard error of approximately 2-3\%, which is significantly smaller than the overall performance variation of nearly 20\% caused by different training configurations. This confirms that the "peaks" and "valleys" are actual features of the optimization landscape rather than the result of random seed variance. As such, Fig. 2 holds as a strong motivation for our approach.
>
> > **For Section 2, I don't think this makes for a comprehensive related work review. The paper introduces BO and scaling laws as core components, but the related works section does not sufficiently discuss prior work using BO and scaling laws specifically for training configuration optimization and performance prediction.**
>
> Thank you for the suggestion. While we only focused on covering prior work on data and model optimization in Sec. 2, we did mention multiple references of various scaling law prediction, joint optimization and BO in the Introduction, Secs. 4.1 and 4.2. We agree that we could have compiled these citations in Sec. 2 for better clarity. We have introduced a new App. G.2 in our revised paper to compile related works on the chicken-and-egg dilemma. Most of these works are related to independent data or model optimization works (which we mentioned throughout our paper), but others cover the literature in performance scaling laws and BO as well.

---

> ### Author Response · Authors · 2025-11-24
> **Response Part 2/3**
>
> > **I am curious why a neural network was chosen as the scaling law predictor. A scaling law is usually a symbolic expression, which makes it explainable and allows it to generalize to unseen extrapolated settings (as the law is not overly complex). A neural network, while a flexible function approximator, can fit the training data well but may not generalize well, is a black-box predictor, and may exhibit non-smooth behavior.**
>
> This is a great question. A family of scaling law formulas [1,2] is usually defined with respect to a fixed training configuration. They do not consider changes in the training configuration (e.g., data mixture, LoRA parameters). Crucially, the scaling of LLM performance varies widely for different training configurations (an example is shown in Fig. 4(d)). For instance, while the scaling curve for one configuration may exhibit exponential growth, a different configuration may show stagnation in performance (again, see Fig. 4(d)). Hence, single scaling law formulas do not have sufficient degrees of freedom to be an effective predictor and we _cannot_ reuse these existing scaling law formulas in our work, where we face a wide range of unknown training configurations during Bayesian Optimization. On the other hand, a neural network predictor, being a *flexible approximator*, is able to model the performance scaling across *multiple configurations* even if they come from a different family of curves. It can also generalize from a few training performance observations to possible unseen training performance curves.
>
> Lastly, Figs. 3 and 6 in our revised paper show that the neural network predictor has a reasonable predictive error. In addition, Bayesian Optimization handles these predictive errors gracefully (Sec. 4.3), outperforming other baselines. Therefore, both our design rationale and the empirical evidence strongly validate our neural network predictor choice.
>
> > **The paper studies "data mixture" as a form of data strategy, parameterizing the data configuration as a probability simplex over N datasets. However, in many real applications, there are no clear "datasets," and data selection is performed instance-wise. How do you see this method handling such a scenario?**
>
> We thank the reviewer for this insightful question. In general, data mixtures [3,4] are _commonly considered in existing literature and real-world settings_, so this is a practical choice in real-world applications.
>
> However, if one encounters a setting where they need to work with instance-wise data, there are several practical strategies to extend our work to more granular-level data selection. In general, directly extending our method to instance-wise data selection increases the complexity of the problem because we end up with a high dimensional integer programming problem on whether to include a particular data instance. This is supported by the high compute time of many existing instance-wise data selection works used in our baselines (Table 4).
>
> To circumvent this problem in practical applications, we can typically group data sources together to form multiple "datasets". For example, chat data from users of different demographics can be consolidated together to form "datasets" for our data mixtures instead of selecting them instance-wise, before applying JoBS over the data mixtures (where different datasets can be optimally upweighted or downweighted). Even when explicit data sources are unavailable, we can use existing semantic models (e.g., ChatGPT's `text-embedding-ada-002` model) to project and cluster instance-level text data based on their semantic similarities. These clusters can be treated as data domains in a data mixture, which then becomes complementary to the JoBS algorithm.

---

> ### Author Response · Authors · 2025-11-24
> **Response Part 3/3**
>
> > **The entire method relies on the Gaussian Process (GP) surrogate being a good model of the true performance landscape $\mathcal{L}$. How good can the GP depict the real metric of $\mathcal{L}$? Although it is computationally expensive to validate this exhaustively, can we get a sense of the surrogate model's fidelity from some smaller-scale experiments (e.g., by comparing the GP's predictions to a densely-sampled ground truth in a low-dimensional version of the problem)?**
>
> We thank the reviewer for their suggestion. In Fig. 1 of our paper, we have illustrated how the true performance landscape is well-approximated by our GP surrogate model (left-side, red) in a smaller, 2-dimensional version of our problem. We hope this provides insight into how well the GP surrogate approximates the true performance landscape.
>
> Beyond just visual inspection, the empirical strengths of JoBS compared to other baselines (Table 3) further substantiate the effectiveness of GP modeling the true performance landscape. If we cannot model the performance landscape well with a GP, then JoBS would not be able to optimize data and model configurations effectively. However, our results show that JoBS *consistently outperforms other baselines* across different model families and different evaluation tasks. As such, this strongly justifies the use of the GP as a good surrogate model.
>
> ___
>
> Once again, thank you for reviewing our paper. We have included additional experiments and analysis in App. G of our revised paper. We hope our responses have answered your questions and improved your opinion of our work.
>
> ___
>
> [1] Kaplan et al. Scaling Laws for Neural Language Models. 2020.
>
> [2] Wu et al. Performance Law of Large Language Models. 2024.
>
> [3] Chen et al. DUET: Optimizing Training Data Mixtures via Feedback from Unseen Evaluation Tasks. 2025
>
> [4] Xia et al. DoReMi: Optimizing Data Mixtures Speeds Up Language Model Pretraining. 2023

---

> > ### Comment · Reviewer_AUVd · 2025-11-26
> >
> > Thanks for the authors’ detailed responses. I still have some reservations about the scaling-law component. In my view, an important merit of scaling laws is that they provide symbolic expressions for prediction, which supports interpretability and potential generalization. There already exist several works on scaling laws for data mixture (e.g., arXiv:2507.09404, 2403.16952) and other training configurations. In such settings, the input variables can be human-designed or even discovered in a way analogous to scientific modeling; they are not fixed a priori. While I agree that existing formulas may not directly fit this problem, I believe it is still worthwhile to explore and potentially discover a new functional form based on the observed data, rather than relying solely on a neural network.
> >
> > Relatedly, I remain somewhat concerned about using a neural network as the predictive tool, given its known vulnerability to adversarial attacks and the uncertainties this may introduce for industrial-scale deployment and reliability.
> >
> > That said, I find the underlying problem interesting and potentially impactful. I will therefore maintain my current score, which reflects a somewhat mixed overall assessment but with a slight lean toward acceptance.

---

> > > ### Author Response · Authors · 2025-12-02
> > >
> > > > ... an important merit of scaling laws is that they provide symbolic expressions for prediction, which supports interpretability and potential generalization. There already exist several works on scaling laws for data mixture (e.g., arXiv:2507.09404, 2403.16952) and other training configurations.
> > >
> > > We agree that symbolic scaling laws offer superior interpretability. However, existing laws typically focus on a single dimension (e.g., data mixture or model size). In our setting, deriving a single functional form that captures the complex, non-linear interaction between the mixed-variable dimensions (continuous data ratios + discrete/continuous model parameters) is computationally intractable. The neural network serves as a flexible and robust predictor for this high-dimensional landscape, though we agree that extending this into a symbolic law is an exciting direction for future work.
> > >
> > > > I remain somewhat concerned about using a neural network as the predictive tool, given its known vulnerability to adversarial attacks and the uncertainties this may introduce for industrial-scale deployment and reliability.
> > >
> > > We thank the reviewer for raising this interesting point. We would like to clarify that our work primarily focuses on the novel problem formulation and our joint optimization solution. As the predictor operates as an internal component within the optimization loop, we consider the vulnerabilities to adverserial attacks to be outside the scope of this paper.
> > >
> > > Thank you again for your review and continued engagement with us. We hope that our clarifications have addressed your concerns and improved your opinion of our work.

---

### Official Review · Reviewer_vLaL · 2025-10-31

**Soundness:** 3
**Presentation:** 3
**Contribution:** 3
**Rating:** 6
**Confidence:** 2

**Summary:**

This paper introduces Joint Bayesian Optimization with Scaling Laws, a framework to co-optimize both training data configurations and model configurations for large language model fine-tuning. The key idea is to treat the fine-tuned model performance as a black-box function over data–model configuration space and use Bayesian Optimization to efficiently search for optimal configurations. To further reduce computational cost, the authors propose a neural performance scaling law predictor that extrapolates the final model performance from short fine-tuning runs. Theoretical analysis provides a cumulative regret bound even under noisy performance predictions. Empirical results on multiple LLMs and tasks demonstrate that JoBS achieves a consistent improvement over independent data or model optimization methods while being faster.

**Strengths:**

The paper focuses on an important question by explicitly formulating the chicken-and-egg dilemma between training data and model configuration in LLM fine-tuning as a joint black-box optimization problem. It proposes JoBS, which combines Bayesian Optimization with a neural performance scaling-law predictor to efficiently explore the configuration space. It models LLM performance as a Gaussian process, uses a predictor to extrapolate final results from short training runs, and provides a theoretical convergence guarantee under prediction noise. Experiments across several LLMs and benchmarks demonstrate consistent performance improvements and faster optimization compared to independent data- or model-selection baselines.

**Weaknesses:**

1. The paper references the term $\gamma_T$ in the main theorem, but does not explicitly define it within the text.

2. All experiments are conducted on relatively small LLMs (up to 8B parameters). It remains unclear whether the proposed method scales to larger backbone models, where optimization dynamics may differ.

3. The compared baselines optimize data and model configurations separately. It would strengthen the empirical evidence to include or discuss any existing methods (if any) that attempt to jointly optimize both components. If JoBS is the first to do so, it should be clearly emphasized.

**Questions:**

Please refer to Weaknesses.

---

> ### Author Response · Authors · 2025-11-24
> **Response Part 1/1**
>
> Thank you for the positive score. We appreciate the reviewer's recognition of the importance of our problem formulation, soundness of JoBS, and the strength of our empirical results. We would like to address the questions raised and hope that our clarifications would improve your opinion of our work.
>
> > **The compared baselines optimize data and model configurations separately. It would strengthen the empirical evidence to include or discuss any existing methods (if any) that attempt to jointly optimize both components. If JoBS is the first to do so, it should be clearly emphasized.**
>
> Thank you for the kind comment and suggestion. As far as we know, JoBS is the *first* to jointly optimize data and model configurations for LLMs. We will make this point clearer in our revised paper.
>
> In addition, our paper has thoroughly discussed existing baselines that only optimize data or model configurations independently (Sec. 5.2). We also, to the best of our ability, adapted existing baselines to consider the interdependence between data and model configurations (and hence provide a fairer comparison to JoBS). These adapted baselines do not work well in practice, as seen in Table 3 -- they either are too computationally expensive (e.g., simply performing BO without performance predictor in Fig.4(a)) or sub-optimal (e.g., alternating data and model optimization methods for a few iterations in Table 3). As such, we believe our baselines provide a fair comparison with JoBS, and corroborate how JoBS is effective as the first algorithm to jointly optimize data and model configurations efficiently. We will include this discussion in our revised paper.
>
>
> > **All experiments are conducted on relatively small LLMs (up to 8B parameters) [...]**
>
> We appreciate the reviewer's concerns regarding the scalability of JoBS to larger models. We have conducted additional experiments to show that JoBS achieves significant performance improvement over existing baselines, even for larger models (specifically Qwen3-14B and Qwen3-32B). We fine-tuned the models on the CommonsenseQA task, and the results show that on average, JoBS achieves a *flat 6\% gain over baselines*, proving its effectiveness on larger models. This is generally consistent with the findings for the smaller Llama-3-8B-Instruct model in our original paper, demonstrating the effectiveness of JoBS even across larger models. We have also included these additional experiments and discussions in App. G of the revised paper (Table 13), and we hope that they have sufficiently addressed your concerns.
>
> |$\downarrow$ Model / Method $\rightarrow$| LESS + AutoLoRA | DoReMi + DARTS | JoBS |
> | :---: | :---: | :---: | :---: |
> |Llama-3-8B-Instruct| 0.80 | 0.79 | **0.84** |
> |Qwen3-14B| 0.82 | 0.80 | **0.86** |
> |Qwen3-32B| 0.83 | 0.84 | **0.90** |
>
> > **The paper references the term $\gamma_T$ in the main theorem, but does not explicitly define it within the text.**
>
> Thank you for pointing this out. $\gamma_T$ represents the *maximum information gain* after $T$ BO observations [1]. We have updated this in our revised paper (Line 287).
>
> ---
> We thank the reviewer for their encouraging feedback and insightful questions. All additional experimental results and discussions have been added to App. G of our revised paper. We hope that our responses have been able to sufficiently address your concerns.
>
> ---
>
> [1] Srinivas et al. Gaussian Process Optimization in the Bandit Setting: No Regret and Experimental Design. 2009.

---

### Official Review · Reviewer_F3SD · 2025-11-01

**Soundness:** 2
**Presentation:** 3
**Contribution:** 2
**Rating:** 2
**Confidence:** 3

**Summary:**

This paper addresses the interdependence between data mixture configuration and model training configuration (LoRA parameters) in PEFT fine-tuning. The authors propose JoBS, combining Bayesian Optimization with a neural network predictor that estimates full fine-tuning performance from short training runs. Experiments on 7 tasks with 3 LLM families show 6-7% improvement over independent optimization and 12.4× speedup. Theorem 4.1 provides convergence guarantees under noisy predictions.

**Strengths:**

**Problem formulation**: Explicitly formulating the interdependence between data mixture ratios and LoRA training configurations as a joint optimization problem is novel. Figure 2b demonstrates that optimal data mixtures vary across LoRA configurations, which is non-intuitive and well-illustrated.

**Technical approach**: Combining BO with a scaling law predictor is sound. Theorem 4.1 shows prediction noise is handled as observation noise. Deep kernels for heteroskedastic modeling and continuous parameterization for mixed discrete-continuous spaces are appropriate. Experimental design with 5-trial averaging and comprehensive baselines is rigorous.

**Empirical validation**: Consistent improvements across tasks and models. Ablations in Figure 4 provide useful insights into component contributions.

**Weaknesses:**

**Severely limited scope**: Only PEFT (specifically LoRA) is evaluated, not full fine-tuning or other PEFT methods (prefix tuning, adapters). The abstract and title should make the scope clear. The claim that "JoBS can also be adapted for LLM pretraining" is unsupported speculation with limited evidence.

**Questionable significance**: If this problem is important, why does no prior work address it? The baseline comparisons require running data mixture optimization then model training configuration optimization separately—no appropriate baseline jointly optimizing both exists, making it unclear if the 6-7% gain comes from joint optimization or simply more compute. The improvement is modest for 9+ GPU hours of optimization.

**Missing practical guidance**: Critical hyperparameters (30 initial samples, Bsmall=50s, 64-width 3-layer MLP) appear arbitrary without sensitivity analysis. No guidance on when joint optimization justifies the computational cost versus using standard LoRA defaults. Scalability to larger models or longer training is unaddressed.

**Lack of failure analysis**: When does the GP smoothness assumption fail? Theorem 4.1 assumes bounded RKHS norm and sub-Gaussian noise but never validates these empirically. Table 3 shows alternating optimization sometimes degrades performance—dismissed as "saddle points" without investigation.

**Questions:**

How sensitive are results to the 30 initial samples, predictor architecture, and Bsmall? Can you provide guidance for practitioners on setting these for new scenarios?

---

> ### Author Response · Authors · 2025-11-24
> **Response Part 1/3**
>
> Thank you for reviewing our paper. We appreciate your comments that our problem formulation is ‘novel’ and our technical approach is ‘sound’, as well as your recognition of our empirical validation and ablations as strengths. We hope that our responses will clarify any remaining doubts, and thank you in advance for engaging us.
>
> > **If this problem is important, why does no prior work address it?**
>
> Thank you for the question. We interpret the “problem” you are referring to as the chicken-and-egg dilemma we described in our paper, which entails the need to *jointly optimize data and model configurations in LLM training/fine-tuning*.
>
> Prior work has optimized model configurations (e.g., [1,2,3]) and data configurations (e.g., [4,5]) independently. However, optimizing these components (i.e., data and model) independently does not guarantee optimal LLM performance [6]. The joint optimization of both components has not been explicitly addressed. We believe this gap exists largely because joint data-model optimization is *considerably more computationally expensive*, and is therefore often handled sub-optimally via heuristics in practice (e.g., fixing the model and tuning the data). Consequently, many would often overlook the possibility of jointly optimizing them due to their perception of the problem being intractable, creating a "chicken-and-egg" dilemma. Therefore, the absence of explicit joint optimization in prior work highlights both the importance of this problem and the novel contribution of our approach.
>
> > **Only PEFT (specifically LoRA) is evaluated, not full fine-tuning or other PEFT methods (prefix tuning, adapters). The abstract and title should make the scope clear.**
>
> Thank you for the comment. It is true that our empirical evaluation focuses on LoRA. This is a strategic choice as LoRA is currently the most widely adopted PEFT method for fine-tuning, and it possesses a sufficiently rich configuration space (rank, $\alpha$, dropout, module-level application, layer positions) to explore the "chicken-and-egg" dilemma. While the JoBS framework is compatible with other PEFT variants (e.g., prefix-tuning), our primary objective is to establish the tractability and "interaction improvement" of the joint optimization process itself. Therefore, rather than broadly evaluating every PEFT variant, we focused our scope on LoRA to provide a clear demonstration of how JoBS resolves the interdependence between data and model configurations. This scope is explicitly stated in our introduction section (Lines 75-76) to ensure the accuracy of our claims, but we will revise the main text to further clarify this distinction.
>
> Nevertheless, to address the reviewer's question of JoBS' applicability to full-parameter fine-tuning, we have *conducted additional experiments on full-parameter fine-tuning and larger model architectures*. We have included these results in App. G of our revised paper. The findings are generally consistent with those observed for LoRA in the original submission, demonstrating the effectiveness of JoBS even when applied to *full-parameter fine-tuning and larger LLMs*.
>
> > **The claim that ”JoBS can also be adapted for LLM pretraining” is unsupported speculation with limited evidence.**
>
> Our statement "JoBS can also be adapted for LLM pretraining” was *written in the conclusion section as the last sentence of the paper*. We think extending JoBS to large-scale LLM pretraining is an important future research direction that is worth pursuing. As such, this statement isn't so much of an "unsupported speculation" but is rather meant to be forward-looking and an important direction for future research.
>
> > **[...] The improvement is modest for 9+ GPU hours of optimization.**
>
> Thank you for the feedback. Many established LLM optimization methods often achieve improvements much less than our flat *6-7\%* improvement across numerous tasks when compared to existing baselines [7]. As such, this performance improvement should be considered significant. Perhaps, the reviewer might have misinterpreted it as a 6-7 *relative* percentage point improvement instead?
>
> Additionally, for clarity, we would like to mention that our flat 6-7\% performance improvement is over the existing data or model optimization baselines, and not over default LLM fine-tuning (which is the most naive approach). If we simply compared our approach to just fine-tuning the LLM (refer to the column marked as default fine-tuning in Table 3), we frequently attain *more than 10\% flat performance improvement*. We hope this clears things up.

---

> ### Author Response · Authors · 2025-11-24
> **Response Part 2/3**
>
> > **The baseline comparisons require running data mixture optimization then model training configuration optimization separately—no appropriate baseline jointly optimizing both exists, making it unclear if the 6-7\% gain comes from joint optimization or simply more compute.**
>
> We appreciate your comment on our baselines. _However, we believe that there is a misunderstanding here._ Please allow us to clarify.
>
> JoBS actually runs *faster or equally fast* compared to other baselines, largely because of our neural network performance predictor which is able to significantly reduce training time. As shown in App. D (Table 4), all data selection (e.g., IF: 52 hours; Diversity: 122 hours) and model selection baselines run slower, or at least take the same duration, as JoBS. Yet, despite the shorter time required by JoBS, it is still able to perform better than the other methods that require a longer runtime. Hence, the 6-7\% gain stems from joint optimization and *not* simply from more compute.
>
> Furthermore, the reported 6-7\% "interaction improvement" is obtained by comparing JoBS against rigorous "mix-and-match" baselines where we combined existing data and model selection methods, rather than using these selection methods in isolation. Since joint optimization is conventionally viewed as computationally intractable as discussed above, a specific baseline that we tried is to apply data and model optimization alternately, which considers the interdependence between data and model configurations like JoBS. However, unlike JoBS, the alternating approach does not co-optimize them and hence does not perform as well.
>
> Therefore, our original submission does already include appropriate baselines to our joint optimization problem and our method, JoBS, incurs *less or the same* compute cost as compared to the other baselines. We hope this helps to clarify the misunderstanding here.
>
> > **Hyperparameters choice (30 initial samples, Bsmall=50s, 64-width 3-layer MLP) appear arbitrary [...] Scalability to larger models [...]**
>
> We appreciate your comment regarding the design choice of our neural network predictor. To clarify, Fig. 4(b) in our original submission provides an ablation study for $B_{\text{small}}$, where we found that a higher $B_{\text{small}}$ allows our predictor to be more accurate, producing better performance. However, choosing a larger $B_{\text{small}}$ incurs a higher computational cost for JoBS. Therefore, we used $B_{\text{small}}=50s$ in our experiments, which is sufficient for JoBS to achieve better performance than other baselines. We will make this point clearer in our revised paper.
>
> We also performed more ablation studies to demonstrate how different design choices surrounding our predictor $\mathcal{F}$ affect its predictions, and how JoBS scales to larger models. The CommonsenseQA task was used for these studies, and the results are included in App. G of our revised paper.
>
> **Ablation on number of initial samples.** In our additional results (Fig. 6 and Table 14), we found that increasing the number of initial samples improves the accuracy of the neural network performance predictor, and thus the performance of JoBS. However, this comes at the expense of a higher computational cost because we need to collect more training samples (and full training runs) initially. For our main experimental results (Table 3), we used 30 training samples across different model families (i.e., Llama, Qwen, Mistral) and tasks, which is sufficient for JoBS to outperform other baselines.
>
> **Scalability of JoBS to larger models.** We also provided additional LoRA-based results (Table 13, also shown below) to showcase how JoBS scales to larger Qwen3-14B and Qwen-32B models under the same experimental setting and hyperparameters as that in the original experiments. Our findings show that JoBS *outperforms existing baselines even for larger models*. This is generally consistent with the results in the original submission, demonstrating the effectiveness of JoBS even when applied to larger models.
>
> |$\downarrow$ Model / Method $\rightarrow$| LESS + AutoLoRA | DoReMi + DARTS | JoBS |
> | :---: | :---: | :---: | :---: |
> |Llama-3-8B-Instruct| 0.80 | 0.79 | **0.84** |
> |Qwen3-14B| 0.82 | 0.80 | **0.86** |
> |Qwen3-32B| 0.83 | 0.84 | **0.90** |
>
> **Effects of varying neural network architecture** In general, when we varied the architecture of our neural network performance predictor, the predictive error did not vary enough to significantly affect the performance of JoBS. Therefore, we used a standard MLP model to represent our predictor. This produces sufficiently good predictions for JoBS to perform well, as we have already shown in Table 3.
>
> We have included these additional ablations and discussions in App. G of our revised paper (Fig. 6, Tables 13 and 14). Please let us know if our additional ablation studies on top of those in the original paper have provided the necessary insights and addressed your questions sufficiently.

---

> ### Author Response · Authors · 2025-11-24
> **Response Part 3/3**
>
> > **When does the GP smoothness assumption fail? Theorem 4.1 assumes bounded RKHS norm and sub-Gaussian noise but never validates these empirically.**
>
> We thank the reviewer for their insightful question. We would like to clarify that our work has explicitly provided empirical validation of the theoretical assumptions made. For instance, we provided evidence (Fig. 1 bottom right, Figs. 2(a) & (b)) to validate the *continuity* property of our performance landscape and show that GP can model it well enough. In addition, we have empirically shown that the predictive error of our performance predictor is *bounded and hence sub-Gaussian* (Fig. 3).
>
> While it is possible that these conditions may not hold in every setting, the assumptions of a bounded RKHS norm and sub-Gaussian noise are standard in the theoretical BO literature when deriving regret bounds [8, 10, 11, 12]. Prior works on BO [13] have also demonstrated that these methods remain highly effective even when the assumptions are strictly violated or when the underlying function is not perfectly smooth. We hope our response has addressed your comment sufficiently.
>
>
> > **Table 3 shows alternating optimization sometimes degrades performance—dismissed as "saddle points" without investigation.**
>
> Thank you for the comment. We would like to clarify what we meant by "saddle points". One of our baselines alternately applied data and model optimization methods to the LLM. This is necessary because existing data optimization methods require us to keep the model configuration choice fixed (and vice versa), so alternately applying the optimization methods has a similar (but not identical) flavour to coordinate descent-style algorithms [9]. When we did that, we observed that, occasionally, the LLM performance becomes worse than before. For instance, when we alternately applied LESS + AutoLoRA to Llama-3-8B-Instruct, the performance dropped from 49% to 46% (this was shown in the paper's result in Table 3).
>
> As a result, this means that alternately applying optimization methods to data and model configurations leads to more undesirable configurations. This is analogous to the phenomenon of "saddle points" observed in some coordinate-descent algorithms [9], where optimization variables might end up, similarly, in undesirable regions of the search space. Therefore, we think this analogy is appropriate in our setting and backed by empirical evidence. We will make this discussion clearer in the revised paper.
>
> ---
> We sincerely thank the reviewer for their detailed feedback and analysis. To address your concerns regarding scope and scalability, we have conducted and included additional experiments on *full-parameter fine-tuning and larger model architectures* in App. G of our revised paper, as mentioned earlier. We hope these results and our detailed clarifications above satisfactorily address your concerns and improve your opinion of our work.
>
> ---
>
> [1] Zhang et al. AutoLoRA: Automatically Tuning Matrix Ranks in Low-Rank Adaptation Based on Meta Learning. 2024
>
> [2] Liu et al. DARTS: Differentiable Architecture Search. 2019.
>
> [3] White et al. BANANAS: Bayesian Optimization with Neural Architectures for Neural Architecture Search. 2020.
>
> [4] Xia et al. LESS: Selecting Influential Data for Targeted Instruction Tuning. 2024
>
> [5] Wang et al. Diversity Measurement and Subset Selection for Instruction Tuning Datasets. 2024
>
> [6] Chen et al. Towards AutoAI: Optimizing a Machine Learning System with Black-box and Differentiable Components. 2024
>
> [7] Wang et al. NICE: Data Selection for Instruction Tuning in LLMs with Non-differentiable Evaluation Metric. 2025
>
> [8] Srinivas et al. Gaussian Process Optimization in the Bandit Setting: No Regret and Experimental Design. 2009.
>
> [9] Wright. Coordinate Descent Algorithms. 2015.
>
> [10] Snoek et al. Practical Bayesian Optimization of Machine Learning Algorithms. 2012.
>
> [11] Wang et al. Recent Advances in Bayesian Optimization. 2022.
>
> [12] Frazier. A Tutorial on Bayesian Optimization. 2018.
>
> [13] Colliandre et al. Bayesian Optimization in Drug Discovery. 2024.

---

> ### Comment · Reviewer_F3SD · 2025-11-24
>
> ```The joint optimization of both components has not been explicitly addressed. We believe this gap exists largely because joint data-model optimization is considerably more computationally expensive, ```
>
> Thanks, this is what I'm looking for.
>
>
> ```
> Nevertheless, to address the reviewer's question of JoBS' applicability to full-parameter fine-tuning, we have conducted additional experiments on full-parameter fine-tuning and larger model architectures. We have included these results in App. G of our revised paper. The findings are generally consistent with those observed for LoRA in the original submission, demonstrating the effectiveness of JoBS even when applied to full-parameter fine-tuning and larger LLMs.
> ```
> Thanks for the new result. As also mentioned by reviewer W1bf21 , I think the claim of the paper can be much stronger with added Full Finetuning results.
>
>
> ```
> To clarify, Fig. 4(b) in our original submission provides an ablation study for
> , where we found that a higher
>  allows our predictor to be more accurate, producing better performance. However, choosing a larger
>  incurs a higher computational cost for JoBS. Therefore, we used
>  in our experiments, which is sufficient for JoBS to achieve better performance than other baselines.
> ```
> Fig. 4(b) seems to compare B=50,100,200, but I can see B=50 has the worst result. Has other hyper-parameter choices been explored. It is puzzling to me why 50 is chosen as for Bsmall.
>
> Also, thanks to authors to add additional ablation on the hyper-parameters.But it does comes as a surprise to me that, according to Figure-6, only 30 initial examples are needed to train predictor. Though adding those ablation gives me more confidence over the generality of the method, there's concern whether waht the predictor actually learned through 30 training example can scale to newer models / architectures.
>
>
> **I have raised my rating of the paper given the clarification and updates.**
>
> Though I still hold reservation regarding the experiment design, where there seems to be no valid baseline that jointly optimizes for both data and model as also pointed out by Reviewer vLaL31. And still, would recommend revise the paper to suggest more research for pre-training, rather than "JoBS can also be adapted for LLM pretraining", without showing supporting experiments.

---

> > ### Author Response · Authors · 2025-11-26
> >
> > Thank you again for the follow-up and for raising our score after the initial review—we truly appreciate it.
> >
> > You raised several remaining questions. **However, after revisiting our initial rebuttal response, we found that these points have already been answered in our earlier rebuttal responses.**
> >
> > ___
> >
> > Specifically, the reviewer pointed out that:
> >
> > > Thanks for the new result. As also mentioned by reviewer W1bf21 , I think the claim of the paper can be much stronger with added Full Finetuning results.
> >
> > Our added results in our initial rebuttal response **are exactly in full-parameter fine-tuning of LLMs across larger LLMs (14B and 32B)**. This is also shown in Table 15 of our revised manuscript.
> >
> > ___
> >
> > In addition, the reviewer mentioned that:
> >
> > > Fig. 4(b) seems to compare B=50,100,200, but I can see B=50 has the worst result. Has other hyper-parameter choices been explored. It is puzzling to me why 50 is chosen as for Bsmall.
> >
> > In our initial rebuttal response, we discussed other hyper-parameter choices (namely on number of training samples, $𝐵_{small}$ and model architecture).
> >
> > Specifically for $𝐵_{small}$, we stated that (copied from our initial response): "Fig. 4(b) in our original submission provides an ablation study for $𝐵_{small}$, where we found that a higher $𝐵_{small}$ allows our predictor to be more accurate, producing better performance. However, choosing a larger $𝐵_{small}$ incurs a higher computational cost for our algorithm (because it implies we need more training time each BO iteration). Therefore, we used $𝐵_{small}=50𝑠$ in our experiments, which is sufficient for JoBS to achieve better performance than other baselines."
> >
> > ___
> >
> > Lastly, the reviewer remarked that:
> >
> > > Also, thanks to authors to add additional ablation on the hyper-parameters.But it does comes as a surprise to me that, according to Figure-6, only 30 initial examples are needed to train predictor. Though adding those ablation gives me more confidence over the generality of the method, there's concern whether waht the predictor actually learned through 30 training example can scale to newer models / architectures.
> >
> > We appreciate that the reviewer find our ablations enlightening and asking the question about the predictor. However, we already answered the above-mentioned question in our initial rebuttal. To summarize:
> >
> > - Figure 6 (App. G.2) demonstrates that the predictor achieves reasonable error across model families and sizes, even for relatively small number of training samples. Our prediction task here is a 1-dimensional output, so the prediction task is not very difficult.
> >
> > - Tables 3, 13, and 15 show that JoBS consistently outperforms baselines even with this predictor.
> >
> > - The Bayesian Optimization framework gracefully handles these prediction errors. Like what you have pointed out in the initial review: "Combining BO with a scaling law predictor is sound (and) Theorem 4.1 shows prediction noise is handled as observation noise."
> >
> > Thus, while the predictor is trained on relatively few examples, both empirical evidence and theoretical guarantees indicate that JoBS remains robust and generalizes well.
> >
> > ___
> >
> > Once again, thank you for reviewing our paper.
> >
> > If there are additional concerns beyond the points listed, we would be grateful if you could let us know, as we are happy to clarify or expand on any aspect that can improve your opinion of our work.

---

> ### Comment · Reviewer_F3SD · 2025-11-26
>
> Thank you for the additional response and the clarifications.
>
> A brief note: **most of my prior response was acknowledging the newly added results, hence increased rating; it is not meant to request more answers.** For points like the choice of Bsmall​, my intent was to probe aspects that remained unclear after reviewing the rebuttal and revised manuscript. I understand the accuracy–compute trade-off, but the specific choice of 50 is still not fully explained—e.g., whether smaller values cause instability or collapse. This remains unanswered.
>
> More broadly, I continue to have concerns regarding (1) the absence of a strong baseline that jointly optimizes both data and model configurations, and (2) how reliably the predictor trained on a small initial set will generalize to future, newer architectures.
>
> Overall, I appreciate the added experiments and the detailed responses, and the points above remain suggestions for tightening and improving the final text.

---

### Author Response · Authors · 2025-12-02
**Rebuttal Summary**

We sincerely thank the AC and the reviewers for their time and effort in providing us with invaluable feedback. We summarize the reviews and our responses below.

**Reviewer F3SD** pointed out that our "problem formulation is novel" and our use of a scaling law predictor with Bayesian Optimization is "sound". **We were able to convince them to raise our score during the rebuttal period**.

* We expanded our evaluation to include full-parameter fine-tuning and larger models of up to 32B (Table 13), demonstrating *consistent performance gains across these settings* in our revised paper.
* We addressed the reviewer's misconception and clarified that JoBS actually *runs faster than or comparable to existing selection baselines while still achieving better performance* (Table 4).
* We provided additional ablation studies for the performance predictor's hyperparameters (e.g., initial sample size, architecture) to empirically justify our design choices (Fig. 6, Table 14).
* Lastly, the reviewer had some queries about our assumptions in our theoretical claims. We highlighted empirical evidence from our original paper (Fig. 1 bottom right, Figs. 2(a) & (b)) that *validates the smoothness of the performance landscape and bounded nature of predictive errors* (Fig. 3) assumed in our paper's theoretical statements.

**Reviewer vLaL** recognized the importance of the problem we are tackling and the strength of our empirical results. In our rebuttals, we made the following clarifications:
* We emphasized that JoBS is the *first work to efficiently co-optimize data and model configurations*, distinguishing it from existing optimization methods that typically focus on a single component and cannot co-optimize both.
* We included new experiments on larger models (Qwen-14B and Qwen-32B), showing that JoBS *is equally effective on larger model sizes and also for full-parameter fine-tuning* (Table 13).

**Reviewer AUVd** appreciated the practicality and motivation of the method, as well as the strong empirical results, and raised questions regarding the predictor choice and budget formulation.
* We justified the use of a neural network predictor over symbolic scaling laws due to the need for flexible approximation across diverse, unknown training configurations.
* We explained why we chose to work with a training time budget as opposed to other budget choice. Specifically, it is *easily interpretable and a constraint commonly faced by practitioners in real-world scenarios*.
* We verified that the performance variance due to PEFT instability is significantly smaller than the optimization gains, validating our preliminary study of the optimization performance landscape.
* We discussed practical strategies for handling instance-wise selection by grouping data into semantic domains to make the problem tractable.

**Reviewer W1bf** acknowledged the novelty of the performance predictor, and asked if JoBS can be extended to larger model sizes and multi-task scenarios.
* We added experiments demonstrating that JoBS effectively optimizes full-parameter fine-tuning for larger models of up to 32B (Table 13), outperforming other baselines.
* We provided additional ablations showing the predictor remains effective and *generalizes well even with lower initial training samples (Fig. 6) and noisy early dynamics (Fig. 5)*.
* We included new results showing JoBS is *capable of optimizing aggregated performance in multi-task scenarios simultaneously* (Table 16).

Once again, we thank the AC and the reviewers for their valuable time and feedback. We truly appreciate your effort and consideration of our work, and hope that the additional experiments and clarifications have strengthened your support for our paper.

Best Wishes,\
Authors

---

### Meta-Review · Area_Chair_FFuF · 2026-01-08

**Summary:**

This paper addresses  the problem in LLM training of jointly optimizing data and model architecture.
For solving this problem, they propose a method that jointly optimizes both
by modeling the performance landscape with Gaussian processes and accelerating search using a performance scaling predictor.

While the proposed framework is conceptually elegant in its attempt
to bridge the gap between data selection and architecture tuning, the empirical validation
and the subsequent rebuttals remains insufficient to  fully convinced about the compromise between
computational overhead and performance gain as well as the applicability of the methods for larger models.
In addition, some concerns remains about the scope of the proposed approach (beyond PEFT) to more
general approach.
As such, I believe that the paper still deserves some revisions and rewriting before meeting an acceptance
level.

**Reviewer Concerns:**

still outstanding:
* scope of the method
* compromise between computational overhead and performance gain

**Reviewer Scores:**

I am not able to answer this.

---

### Decision · Program_Chairs · 2026-01-26

Reject